behaviour/cognition

cooperation, instrumental helping, altruism, chimpanzees, bonobos

**Author for correspondence:**
Suska Nolte
e-mail: suska_nolte@eva.mpg.de

# Targeted helping and cooperation in zoo-living chimpanzees and bonobos

Suska Nolte[1,2] and Josep Call[1,2]

[1]School of Psychology and Neuroscience, University of St Andrews, St Andrews, Fife, UK
[2]Developmental and Comparative Psychology, Max Planck Institute for Evolutionary Anthropology, Leipzig, Sachsen, Germany

SN, 0000-0002-2614-792X

Directly comparing the prosocial behaviour of our two closest living relatives, bonobos and chimpanzees, is essential to deepening our understanding of the evolution of human prosociality. We examined whether helpers of six dyads of chimpanzees and bonobos transferred tools to a conspecific. In the experiment 'Helping', transferring a tool did not benefit the helper, while in the experiment 'Cooperation', the helper only obtained a reward by transferring the correct tool. Chimpanzees did not share tools with conspecifics in either experiment, except for a mother–daughter pair, where the mother shared a tool twice in the experiment 'Helping'. By contrast, all female–female bonobo dyads sometimes transferred a tool even without benefit. When helpers received an incentive, we found consistent transfers in all female–female bonobo dyads but none in male–female dyads. Even though reaching by the bonobo receivers increased the likelihood that a transfer occurred, we found no significant species difference in whether receivers reached to obtain tools. Thus, receivers' behaviour did not explain the lack of transfers from chimpanzee helpers. This study supports the notion that bonobos might have a greater ability to understand social problems and the collaborative nature of such tasks.

## 1. Introduction

Many animal species live in complex social groups, where individuals compete over resources and also show a variety of seemingly complex cooperative behaviours. The term cooperation has been defined as actions that are beneficial to both the actor and the recipient [1]. In order for a cooperative act to occur, both individuals need to invest into reaching a mutual goal and both consequently gain a benefit from it. We find cooperative behaviours in various distantly related species such as ants [2], cleaner fish [3], ravens [4], wolves [5] and

elephants [6]. The ability to help and cooperate with others, therefore, seems to have evolved convergently across taxa [7]. Gauging differences in the expression of cooperative behaviours between closely related species and relevant socio-ecological factors enables the assessment of the factors that could have played a role in the emergence of cooperative behaviours. In the current paper, we thus set out to understand how two closely related species, the chimpanzee (*Pan troglodytes*) and bonobo (*Pan paniscus*), differ with regard to their capacity to cooperate and help conspecifics.

Wild chimpanzees have been observed to go on boundary patrols together, form coalitions to fight against other males in the group or invading outside groups, hunt collectively and share meat thereafter (e.g. [8–10]). Moreover, chimpanzees produce alarm calls to inform others of danger [11]. Most of these behaviours such as boundary patrols, coalition formation, and collective hunting and meat-sharing occur between male chimpanzees [8–10,12]. It is yet unknown to what degree actions such as consolation and meat-sharing are prosocially motivated, and hence are done with the intent of the action to benefit the other [1], or to what extent they understand the role of their partner during collective behaviours.

In experimental contexts, chimpanzees do not seem to discriminate between options in which their partner either does or does not receive a reward (e.g. [13,14]; but see [15]). Tan *et al.* [16] argued that such 'prosocial choice paradigms' might inherently be too difficult to understand as even children sometimes fail to understand the task's contingencies. For example, while children acted prosocially when this came at a cost to them, they did not when it was cost-free [17]. The researchers suggested that children did not understand the task's contingencies, which supports the notion that slight differences in design can lead to outcomes of seemingly unprosocial individuals [17]. This view is further supported by a recent study, in which children acted prosocially when facing each other during the test instead of side-by-side, as was done in previous studies [18]. Therefore, the 'prosocial choice paradigm' in itself might be easily affected by changes in the set-up that may create false negatives. Studies that used an 'instrumental helping paradigm' have been more consistent in showing that chimpanzees seem willing to help another conspecific achieve direct goals without benefiting from such help. For example, Warneken & Tomasello [19] found that chimpanzees released a peg that unlocked a door, enabling a conspecific to enter an adjacent room and consume food located there. Similarly, chimpanzees provided access to rewards by releasing a peg that held a baited apparatus out of reach from a conspecific [20]. Finally, Yamamoto *et al.* [21,22] reported that chimpanzees transferred objects to a conspecific which allowed her to rake in food that was placed out of reach (but see [23]). Such results seem to suggest that chimpanzees behave in a prosocial manner towards conspecifics, and behaviours observed in wild populations could have such underlying intentions.

In addition to the design used to test prosociality, another potential factor that could influence whether chimpanzees act prosocially is whether the partner draws attention to herself or the task [20]. It seems that prosocial behaviour virtually disappears if the individual in need does not signal her need, for instance, by stretching her arm into the subject's cage to reach for the needed object or shaking a chain connected to the apparatus (e.g. [20–22]). Some authors have interpreted the signalling need as a form of social harassment [24]. According to this interpretation, food donors may relinquish food to avoid social harassment in the form of begging behaviour rather than as an indication of a purely altruistic motivation. Others proposed that signalling behaviours might further facilitate understanding of the task as the partner's goals are made more salient [20]. A third explanation is that signalling behaviours could elicit responses not because they constitute harassment or facilitate understanding the partner's goal but because they call attention to the task at hand [25,26]. Thus, in this sense, signalling behaviours would act as mere stimulus or local enhancement. Tennie *et al.* [25] adapted the design used by Melis *et al.* [20] and introduced an additional condition, in which pulling the peg hindered instead of granted access to the apparatus and its rewards. Thus, they tried to determine whether prosocial or spiteful intentions might underlie peg releases and predicted to find no difference between both conditions because signalling behaviours would lead to peg releases in both conditions. Indeed, they found that manipulation rates were the same irrespective of the outcome for the partner. The authors suggested that such actions were, therefore, most likely done because of stimulus enhancement and not because of either prosocial or spiteful underlying motivations. These results call into question the interpretation of previous results, in which chimpanzees had to manipulate an object to help the partner (e.g. [19,20]). Even though some results (e.g. [21,22]) cannot be explained by mere stimulus enhancement as the objects could not be touched by the recipient, it is important to tease apart whether prosocial intentions were underlying such actions.

Though bonobos and chimpanzees are similarly closely related to humans, chimpanzees are often used as the representative of the great ape family and our closest living relative, neglecting the differences between species. This renders the conclusions drawn about the evolution of human prosocial abilities biased towards the behaviour of a single species.

In contrast to chimpanzees, there is little evidence that bonobos hunt together in the wild, go on collective boundary patrols or fight outside groups [27–29]. Similar to chimpanzees, bonobos form coalitions and support each other during in-group fights [30]. Moreover, both species have been observed to share food with other adult group members [9,10,31,32]. In chimpanzees, coalition formation and food sharing in the wild mainly occurs between adult males, though meat-sharing between males and females has been reported [9,10]. In bonobos, however, the strongest coalitions and food transfers among adults nearly exclusively occur between females [30–32]. Given that females migrate to other groups after adolescence in both species, coalition formation and food sharing occurs irrespective of kinship in bonobos [30]. This pattern is also reflected in the fact that while chimpanzees tend to associate most strongly with same-sex partners, male bonobos associate most strongly with female bonobos (i.e. their mothers) and female bonobos with other females [33]. Surbeck et al. [33] argue that such association patterns might reflect the tactical selection of potentially best cooperation partners. It might, therefore, explain high male–male cooperation levels in chimpanzees and increased mating success for bonobo males with strong associations with their mothers [33]. Thus, association patterns within a group might be used as a proxy to potentially infer who the best cooperative partners are.

Differences between bonobos and chimpanzees have also been documented at the neural level. Compared to chimpanzees, bonobos have more grey matter in the right anterior insula, the amygdala, and a pathway linking the amygdala and the ventral medial prefrontal cortex [34]. In humans, these areas are part of the visceral brain centres and involved in heightened autonomic reactivity to emotional stimuli [35], supporting socio-emotional processing and possibly empathy through emotional contagion and interoceptive abilities [36]. Herrmann et al. [37] found that bonobos performed better than chimpanzees on tasks related to theory of mind abilities, possibly reflecting the differences found in brain areas. Clay et al. [38] showed that bonobos are sensitive to whether social expectations during aggressive conflicts are violated. Screams emitted by victims of aggression varied depending on whether the aggression was unexpected or could be socially predicted (e.g. during resource competition). Even though more research is needed to understand the motivation to emit such distinct screams, bonobos seem to possess social expectations based on the situation they are in and what events preceded it. Using eye-tracking, Kano et al. [39] found that bonobos looked at the face and eyes of conspecifics longer than chimpanzees did. This finding further supports the idea that bonobos might be more motivated, and possibly, more sensitive than chimpanzees to attend to social cues.

One theory that has been proposed to explain the heightened sociality of bonobos compared to chimpanzees is their feeding ecology (e.g. [40,41]). Across chimpanzee groups, the proportion of time that females spend together while searching for food varies greatly depending on fruit availability [42]. Given that bonobo habitats have larger and more stable food availability [43–45], females tend to remain in larger social groups throughout the year and, unlike chimpanzee females [42], are rarely seen alone [46]. Additionally, food availability affects the females' health and thus the frequency and length of their fertile cycles [47]. This in turn reduces the number of males per fertile female in a group, which increases female mating choice and possibly decreases male aggression due to mating competition and reduced options of mate-guarding [29,48–50]. In combination with increased female sociality, increased mate choice might have allowed bonobo females to select less aggressive males and form coalitions against those that are [41]. Differences between chimpanzee communities with regard to male aggression levels and female gregariousness [28,42,51] support the notion that feeding ecology immediately influences social behaviour. However, as Jaeggi et al. [50] pointed out, research needs to formally address the impact of resource defensibility and demographic factors on chimpanzee aggression levels.

Hare et al. [41] proposed that the factors mentioned above led to bonobo self-domestication; hence, selection against male aggression that resulted in several morphological, behavioural and neurological differences compared to chimpanzees. The authors argued that as female groups punish males with high aggression levels, males that are less aggressive towards females and other males will gain fitness benefits through alliances and increased reproduction. This influences genes that affect the development of adult aggression, and simultaneously, influences attributes that are controlled by the same genes as a by-product, e.g. altered morphological traits [41]. Differences between bonobo and chimpanzee males can also be found at the neurological and hormonal level. For example, Wobber et al. [52] showed that when anticipating competition, bonobo males release cortisol, whereas chimpanzee males release testosterone. In humans, increased testosterone levels have been associated with a higher power motive, while increased cortisol levels with a passive coping-style during competition [53]. Genetic and hormonal differences possibly result in a juvenilization (i.e. retention of juvenile behaviour in adulthood) of several behaviours such as increased tolerance, play and

emotional reactivity [41]. Thus, the higher sociality of bonobo females and reduced aggression of bonobo males compared to chimpanzees might have allowed the development of heightened awareness to social cues, through which bonobos possibly became more adept at processing socially relevant stimuli. However, we do not yet know whether this heightened awareness and cognitive ability translates into more altruistic responses in bonobos compared to chimpanzees.

So far, only a handful of experimental studies directly compared the overt behaviour of the two species. Most studies focused on either species, making it difficult to assess if differences might be due to methodological rather than species differences. In one of the few studies that have directly compared cooperation in both species, Hare *et al.* [54] found that bonobos cooperated significantly more than chimpanzees when resources were clumped, thereby increasing the likelihood of a conflict. By contrast, both species cooperated at a similar level when resources were dispersed. The researchers argued that bonobos' greater level of social tolerance might have been responsible for this result. Similarly, bonobos transferred more tokens to conspecifics than chimpanzees in a token exchange paradigm [55]. These two studies suggest that bonobos are more adept at cooperating with one another than chimpanzees. Three additional studies found that neither species acted prosocially towards a conspecific: neither bonobos nor chimpanzees transferred tools to a partner in need [23], shared food with a conspecific by opening a door into their own cage [56] or chose a prosocial option more often than a selfish one [14]. In contrast with the two studies that found bonobos to be more tolerant or cooperative [54,55], Jaeggi *et al.* [40] argued that in fact chimpanzees were more prosocial than bonobos as they observed chimpanzees to transfer food more tolerantly and proactively than bonobos. Similarly, Cronin *et al.* [57] contrasted social tolerance measures of bonobos to those previously obtained from chimpanzees [58], and found that chimpanzees were on average more socially tolerant during feeding events than bonobos. Thus, the current mixed results require experiments that directly compare the two species while controlling for motivational aspects, given that it is still unclear which motivation underlies chimpanzees' (and for that matter also bonobos') prosocial behaviours.

We tested both species with the same methods and directly compared their responses. We presented six chimpanzee and six bonobo dyads with an instrumental helping task, in which the helper did not gain any benefit by helping, and a cooperative task, in which both partners gained rewards through cooperation. In both tasks, one individual (helper) was given access to tools while the partner (receiver) could only operate her apparatus and access the rewards, provided the helper transferred the correct tool. The only difference between the two tasks was whether the helper had a direct benefit from transferring the tool or not. In case the helper transferred no tool or an incorrect tool, the receiver could not operate her side of the apparatus and the rewards remained blocked for both individuals. This set-up gave us a unique opportunity to answer several questions while controlling for methodological artefacts. First, we investigated whether we find a species difference between bonobos and chimpanzees in their ability to solve cooperative tasks and motivation to behave altruistically. As mentioned earlier, only a handful of studies included both species and assessed either cooperation or altruism, leading to conflicting evidence when comparing findings across studies (e.g. [40,54,55]). Second, we assessed the motivational aspects (self-regarding versus other regarding preferences) underlying prosocial behaviour. Thus, we studied how the apes responded to their partner's needs depending on whether they also benefited from helping them. Third, we examined the cognitive complexity underlying helping and cooperation, i.e. whether the helper was able to distinguish between her own and the other's needs in terms of the tools required to solve the task. Depending on the condition, both apes needed to use the same tool type or different tools. In the latter case, the helper had to perform a self–other distinction and understand that the partner needed a different tool to the one she needed herself. Human children become increasingly able to understand that partners' needs might be diverging from their own only around the age of 2 years [59], and to our knowledge, this is the first experiment specifically addressing the question of whether non-human great apes can demonstrate such a self–other distinction during helping or cooperating.

## 2. Methods

### 2.1. Subjects

We tested six bonobos (5 females, $M_{age} = 12.04$) and six chimpanzees (3 females, $M_{age} = 17.13$) housed at the Wolfgang Koehler Primate Research Center in Zoo Leipzig, Germany (table 1; electronic supplementary material for details). Dyads were assigned following a round-robin design per species,

**Table 1.** Dyads tested.

| species | name helper | name receiver | sex helper | sex receiver | age helper | age receiver |
|---|---|---|---|---|---|---|
| bonobo | Fimi | Gemena | F | F | 5.76 | 8.48 |
| bonobo | Gemena | Luiza | F | F | 8.48 | 9.26 |
| bonobo | Luiza | Kuno | F | M | 9.26 | 17.43 |
| bonobo | Kuno | Lexi | M | F | 17.43 | 15.39 |
| bonobo | Lexi | Yasa | F | F | 15.39 | 16.68 |
| bonobo | Yasa | Fimi | F | F | 16.68 | 5.76 |
| chimpanzee | Fraukje | Kara | F | F | 38.58 | 9.53 |
| chimpanzee | Kara | Kofi | F | M | 9.53 | 9.49 |
| chimpanzee | Kofi | Lobo | M | M | 9.49 | 10.70 |
| chimpanzee | Lobo | Lome | M | M | 10.70 | 13.15 |
| chimpanzee | Lome | Sandra | M | F | 13.15 | 21.33 |
| chimpanzee | Sandra | Fraukje | F | F | 21.33 | 38.58 |

so that each ape was included in two dyads. We selected chimpanzee and bonobo dyads based on three criteria: (i) reliable participation in previous tests of cooperation, (ii) tolerance between members of the dyad, and (iii) positive relationship between them based on the zoo keepers' experience. For logistic reasons, we could only include four female–female and two mixed-sex bonobo dyads as two additional adult bonobo males could not be tested (total group size: 10, incl. 2 infants). We included six chimpanzees to match the number of bonobos, and tested two female–female, two male–male and two mixed-sex chimpanzee dyads (total group size: 18, incl. 1 infant). Both species were housed in their respective social group in an indoor enclosure and during summer in an additional outdoor enclosure. All individuals had previously participated in cognitive tests and were familiar with the handling procedure during tests. At no point was any ape food-deprived and water was provided ad libitum.

## 2.2. Apparatuses

In the experiment 'Helping', we used two distinct apparatuses that could each be operated by a respective tool. In the experiment 'Cooperation', we used four apparatuses that were a combination of the original two from the helping experiment.

The first apparatus we used in the experiment 'Helping' (hereafter, 'apparatus Stick'; figure 1a) could be operated by inserting a wooden stick (25 cm length, 0.6 cm in diameter) into a small tube at the top of the apparatus. Pushing the stick forward displaced a small bottomless container baited with grapes. As soon as the container reached the location of a hole in the platform, the grapes fell onto a slide that led towards the cage's mesh (i.e. the feeding area) where the ape could retrieve and consume them. The stick remained available to the ape throughout the entire session, as it was easily retrievable from the apparatus by pulling it out of the tube. The second apparatus in the experiment 'Helping' (hereafter, 'apparatus Block'; figure 1b) could be operated by inserting a solid plastic cuboid (3.5 cm × 3.5 cm × 3.5 cm) into a round hole at the top of the apparatus. Dropping the block through the hole caused a lever to tilt by the block's weight. As the lever tilted, a string was pulled upward that in turn lifted a plastic barrier, thus releasing the grapes placed behind the barrier. Upon release, the grapes rolled down the slide and towards the mesh of the ape's cage (i.e. feeding area). As with the stick, the block remained available throughout the entire session because as soon as the lever was tilted the block dropped down to the feeding area.

In the experiment 'Cooperation', we used four distinct apparatuses that were a combination of the two original ones and could be operated with the same tools. These apparatuses were built to form a 90° angle, so that both sides could be attached to the respective mesh of two adjacent cages and operated simultaneously (see electronic supplementary material, for pictures). Instead of using only one tool, the apes now had to use two tools to operate any of the four apparatuses: either two sticks for both sides (apparatus Stick-Stick), two blocks for both sides (apparatus Block-Block), or one stick for one side and one block for the other side (apparatus Stick-Block) and vice versa (apparatus Block-

(*a*)                                    (*b*)

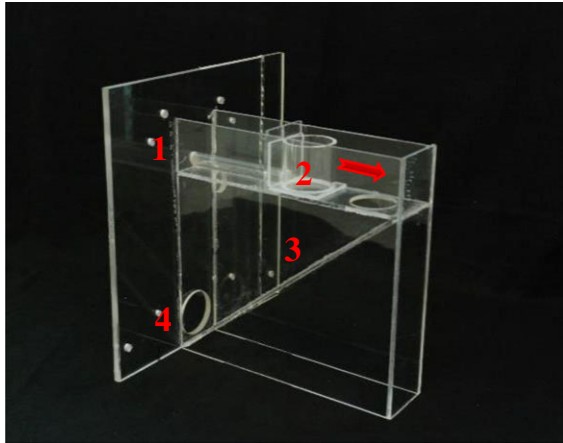
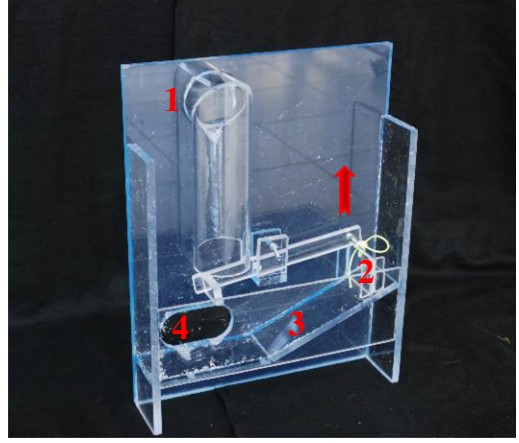

**Figure 1.** Picture of the 'apparatus Stick' (*a*) and 'apparatus Block' (*b*). The flat surface was attached to the testing cage and faced the ape. Red numbers indicate the location where the tool needed to be inserted (1), the location of the grapes (2), the slide (3) and the feeding area (4). The red error, respectively, indicates the direction of movement of the container or lever.

Stick). The general mechanism was the same for each apparatus; two platforms hindered the rewards from falling down onto a slide. One platform could be released by operating one side of the apparatus and the other platform could be released by operating the other side of the apparatus. In case the left side was operated first, the rewards fell onto the next lower platform and stayed there until the right side was operated as well. In case the right side was operated first, the rewards stayed on the upper left platform but fell down immediately once that platform was released since the lower right platform had already been released. This way, it did not matter which platform was released first and the rewards stayed in place until both sides were operated. After successfully operating both sides, five grapes fell down onto each of two slides and rolled towards the respective mesh of both cages where the corresponding ape could consume them (see electronic supplementary material, for detailed description of each apparatus). Black tape was wrapped around the tube (sides that needed to be operated by a stick) and platform (sides that needed to be operated by a block) to highlight the main features and make it easier for the apes to assess which tool needed to be used.

## 2.3. Design

We conducted two different experiments consecutively presented to each dyad (figure 2). We started with the experiment 'Helping' (hereafter, 'Helping 1'), in which tool transfers had no direct benefit for the helper. Subsequently, we presented each dyad with experiment 'Cooperation', in which transfers led to a direct benefit for the helper. To rule out order effects, we switched back to experiment 'Helping' (hereafter, 'Helping 2') for all dyads that transferred a tool at least once in experiment 'Cooperation', resulting in an ABA design for these dyads (figure 3). Before each experiment, apes received individual training to ensure that (i) they understood the apparatuses' mechanisms and (ii) helpers knew whether they could work independently of the receiver or not (see electronic supplementary material, for full training protocol). During training, none of the apes was paired with a conspecific and never had to transfer a tool to a different cage in order to retrieve the rewards. The training merely consisted of familiarizing the individuals in how to use the tools on each apparatus. During training of the experiment 'Cooperation', the sliding door connecting the two cages was open, which allowed each ape to operate both sides of the apparatus alone. Testing took place in the sleeping rooms of the respective groups. The left cage, which was occupied by the helper, was 2.7 × 2.5 m wide and the right cage, occupied by the receiver, was 3.2 × 3.9 (chimpanzees) and 2.9 × 3.9 (bonobos) m wide (see electronic supplementary material, for detailed measurements).

To assess if helpers that did not transfer tools in experiment 'Cooperation' failed to understand that both sides needed to be operated, we implemented a knowledge control that adhered to the same procedure as the training of experiment 'Cooperation' (figure 3). All apes demonstrated that they understood the mechanism (see electronic supplementary material, for details).

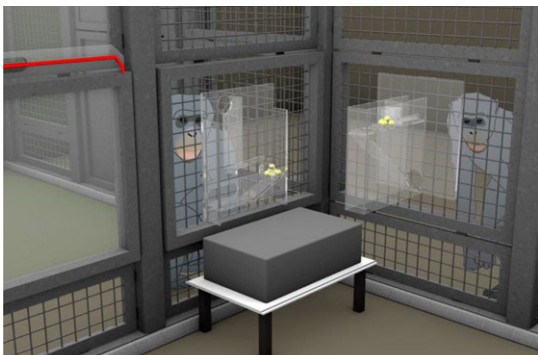
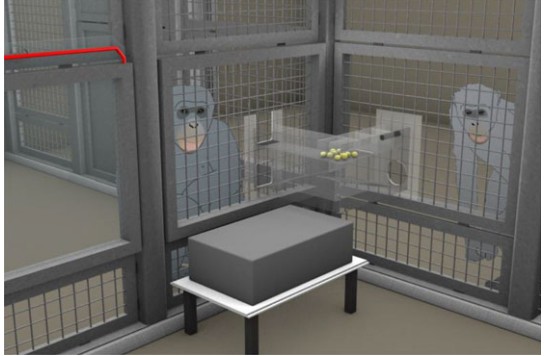

**Figure 2.** Set-up of the experiment 'Helping' (*a*) and experiment 'Cooperation' (*b*), depicting the condition 'Different'. The same set-up was used in the condition 'Same' and the control condition. The tools were located underneath the grey box and a second box was used in the control condition.

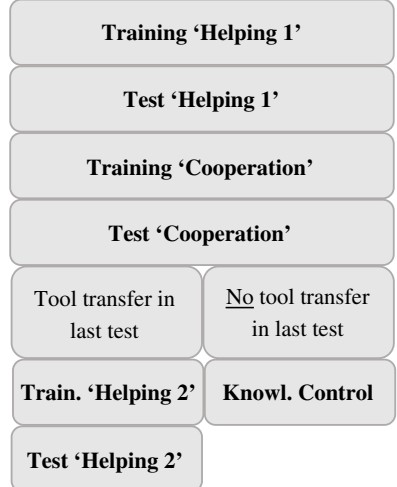

**Figure 3.** Flowchart of the entire study design.

## 2.4. Procedure

We tested each ape with two different partners. In one dyad, the ape was assigned the role of the helper and in the other dyad the role of the receiver (table 1). During the test, the individuals of a pair sat in adjacent cages with the sliding door closed. The mesh separating the two cages provided good visibility into the conspecific's cage and allowed for any interactive behaviours and the transfer of tools. In the experiment 'Helping', we baited both apparatuses with five grapes each. In the experiment 'Cooperation', we baited each apparatus with 10 grapes, five of which could roll towards the helper and five towards the receiver. Even though both individuals needed a tool, only the helper received access to an assortment of tools. Thus, the helper needed to share a tool with the receiver in order for the receiver to operate her apparatus (experiment 'Helping') or her side of the connected apparatus (experiment 'Cooperation'). In the experiment 'Helping', the apparatuses could be operated separately, which means that the helper could retrieve her own rewards independent of whether the receiver was able to access her rewards. Given that in the experiment 'Cooperation', both sides had to be operated to release the rewards, the helper now needed to share a tool with the receiver in order for both individuals to receive any rewards. This created an incentive for the helper to share a tool with the receiver and was the main difference to experiment 'Helping'. The helper always received access to an assortment of three tools. The assortment corresponded to the type of apparatus presented to the helper, which meant we gave two tools of the type that she needed on her side and one tool of the other type. Even though the tool she needed was retrievable from the apparatus after using it, we included two exemplars of this type just in case they might have attributed more value to

a tool they used than to one they did not need. Therefore, we offered an additional tool, so there was always a spare tool of the type they had to use themselves.

We used three conditions in both experiments. In the *condition 'Same'*, both conspecifics were presented with the same type of apparatus, hence in the experiment 'Helping', both received either a duplicate of 'apparatus Stick' or 'apparatus Block' and in the experiment 'Cooperation', they received 'apparatus Stick-Stick' or 'apparatus Block-Block'. Given that the assortment of tools depended on the type of apparatus presented to the helper, the chance of randomly transferring the tool needed by the receiver was 2/3 since both needed the same tool. In the *condition 'Different'*, both conspecifics received the opposite type of apparatus. Thus, in experiment 'Helping', one conspecific faced 'apparatus Stick' and the other 'apparatus Block', or vice versa, and in experiment 'Cooperation', they received 'apparatus Block-Stick' and 'apparatus Stick-Block'. The chance of randomly transferring the tool needed by the receiver was 1/3 since both needed different tools. In the *control condition*, we presented each of the four different scenarios that we used in the test conditions. The only difference between the two test conditions was that the receiver was now also given access to a separate assortment of tools, where the combination of tools was dependent on the type of apparatus presented to the receiver. Therefore, there was no need for the helper to share a tool with the receiver as she could access her apparatus or side of the apparatus with the tools given to her. Dyads received 12 trials in each condition, with one trial of each of the three conditions per testing day. Thus, on day 1, they received one trial of condition 'Same', one trial of condition 'Different' and one trial of the control condition—resulting in 12 testing days per dyad and experiment. The order in which we presented the conditions each day was randomized and bonobo dyads received the same testing schedule as chimpanzee dyads. Each trial lasted 3 min regardless of whether the respective helper transferred a tool or not.

## 2.5. Data coding

We coded whether tool transfers occurred, the order in which each tool type was transferred during a given session, and four distinct behaviours that could have been potentially exhibited by the receiver. We scored for each second of a given trial whether any of the behaviours occurred. Transfers were considered when a tool initially given to the helper was directly transferred to the cage of the receiver by the helper, or when the receiver was allowed to grab a tool close to the mesh (such tolerated theft occurred in two cases). To assess the level of negative arousal, we coded the occurrence of scratching and banging (e.g. [60]). We defined scratching as 'rake one's own hair or skin with fingernails including large movements of arm' ([61, p. 1036]) and banging as using either hands or feet to hit the mesh separating the recipient from the helper or the mesh onto which the apparatus was mounted. To understand whether receivers made an effort to acquire tools from the helper, we coded the occurrence of reaching. Reaching was operationalized as putting at least one finger through the mesh, which separated the cages of two conspecifics. Finally, we included maintaining close proximity as a measure of whether the receiver tried to stay close to the helper. We operationalized close proximity as remaining in a maximum distance of 50 cm to the mesh separating the two conspecifics. In case requesting gestures were exhibited only rarely, this measure enabled us to understand if the receiver did not gesture even though she was attentive to the helper's actions or if she did not regard the helper at all.

To assess interrater reliability, we randomly selected 20% of the bonobo data (59 sessions) that a research assistant coded blind to the procedure and hypotheses. We selected the bonobo data only given that nearly all models were based on these data. To assess the four behavioural variables (reaching, close proximity, scratching and banging), we used Cohen's $\kappa$ to evaluate whether the two raters agreed on the occurrence for each of the 180 s of a session. We acquired sufficient and good reliability for each of the four behavioural variables (Reaching: $K = 0.72$; Close proximity: $K = 0.93$; Scratching: $K = 0.80$; Banging: $K = 0.73$). Additionally, we assessed whether the two raters agreed on the type of tool that (if any) subjects shared during a session. To do so, both raters noted down the type and order of transferred tools. Again we acquired good reliability between the two raters ($K = 0.89$). For subsequent analyses, we used the data of the original coder.

## 2.6. Analyses

We analysed the data from three different perspectives: (i) what factors influenced the likelihood that helpers transferred tools, (ii) whether helpers were able to tailor their prosocial acts according to the

needs of the receiver, (iii) how the receivers behaved depending on the species they belonged to and when receiving the correct tool.

## 2.7. Tool transfers

To assess the factors that influenced tool transfers, we based the main analyses only on the data of the bonobos, since across all sessions and dyads, chimpanzees only transferred a tool twice. We fitted a Generalized Linear Mixed Model (GLMM; [62]) with a binomial error structure and logit link function [63] and used the occurrence of at least one transfer (Yes, No) in a given session as our response variable. The number of observations was 656 of six dyads. To understand whether it made a difference if helpers benefited from transferring a tool and whether order effects were present, we used the type of experiment with three levels ('Helping 1', 'Cooperation', 'Helping 2') as one of our predictors. Additionally, we included condition with three levels ('Same', 'Different', 'Control') to understand whether helpers transferred tools irrespective of whether the receiver actually needed a tool. We included the $z$-transformed predictor session number to assess whether helpers were more likely to transfer tools at the beginning, middle or end of the experiment indicating either motivational issues or time needed to understand the task. To understand how this factor interacted with the predictor experiment, we further included an interaction term of these two factors. This interaction term informs about whether motivational or learning factors influenced the helpers to different degrees depending on whether they benefited versus not benefited from helping. Finally, we included the predictor reaching with two levels (Yes, No) to understand how reaching by the receiver influenced the probability of a transfer. To control for any influences that the age of the receiver and helper together might have had on the likelihood of transfers (i.e. older individuals might have shared more tools when the receiver was young), we included an interaction term between the $z$-transformed variables age of the helper and age of the receiver, and their two respective main effects. Finally, to keep type 1 error rates at the nominal level of 5%, we also included the random intercepts [64,65] for test day and dyad identity, and the random slopes components [64,65] within dyad identity for the fixed-effects experiment, condition, reaching, session number, the interaction between experiment and session number, age of the helper, age of the receiver, and the interaction between age of the helper and age of the receiver.

## 2.8. Tool selection

We further assessed how bonobo helpers selected the first tool that they transferred in order to understand whether the tools were selected according to the others' needs. We fitted a GLMM [62] with a binomial error structure and logit link function [63] and our response variable was whether the first transfer in a given session, in which a transfer occurred, was correct and had two levels (Yes, No). Therefore, only the first transfer in a given session was considered and all second or third transfers were discarded. The number of observations was 135 of four dyads. To assess whether the type of experiment influenced the occurrence of correct first transfers, we included the predictor experiment with three levels ('Helping 1', 'Cooperation', 'Helping 2'). This was done to understand whether helpers might have paid more attention to the needs of the receiver when they themselves benefited from transferring the correct tool (i.e. in experiment 'Cooperation'). We included the predictor condition with two levels ('Same', 'Different') to understand whether one of the two conditions might have been easier for the helper, for example, if they both needed the same tool. We used a log-transformed offset term [63] to account for the fact that the chance of transferring the correct tool was different between the two conditions since the helper always received two tools of the type she needed herself and one of the remaining type. Thus, in the condition 'Same', in which both needed the same tool, the chance of transferring the correct tool was 2/3. In comparison, the chance of transferring the correct tool in the condition 'Different' was 1/3 since the receiver needed a different tool than the helper. Finally, we included a predictor specifying which tool the receiver needed in a given session with two levels ('Stick', 'Block') to understand whether the helper was preferably transferring a specific type of tool. To control for any learning effects that might have influenced the ability to transfer the correct tool, we included the $z$-transformed variable session number. Additionally, to control for any influences that the age of the receiver and helper together might have had on the likelihood that the helper shared a tool (e.g. young individuals might have been more likely to receive tools by old individuals), we included the interaction term between the $z$-transformed variables age of the helper and age of the receiver, and the two respective main effects.

Finally, to keep type 1 error rates at the nominal level of 5%, we also included the random intercepts [64,65] for test day and dyad identity, and the random slopes components [64,65] within dyad identity for the fixed-effects experiment, condition, tool of the receiver, session number and age of the helper.

Given that all the key predictors of the model had a non-significant influence on the response, we continued to manually dummy code and then centre each categorical predictor. We then calculated two new GLMMs while maintaining the same structure as in the initial model, except that in one model, we used the condition 'Same' as a reference and in the other the condition 'Different'. Further, we used an intercept optimization (the R-function was written by Roger Mundry and is available upon request) on these two new models. The resulting intercept was assessed to gauge the average probability of correct first transfers while controlling for the effect of each predictor in the model. Finally, the respective intercepts were tested against a chance level of 2/3 (condition 'Same') and 1/3 (condition 'Different') to understand if transfers were done randomly in one or both test conditions while controlling the influences of the other factors in the model.

## 2.9. Receivers' behaviours

We first assessed whether the two species differed in terms of the behaviour executed by the receivers and if this might have explained the difference in transfers done by the helpers. Given that chimpanzees shared close to no tools but bonobos already shared substantially more tools in experiment 'Helping 1', frustration might have already influenced chimpanzee receivers differently than bonobo receivers in experiment 'Cooperation'. Therefore, we only focused on the data from experiment 'Helping 1' as this seemed to be most comparable.

We initially fitted linear mixed models [62] with a Gaussian error structure and identity link function, separately for each behaviour (i.e. scratching, banging, requesting and close proximity) expressed by the receivers. However, none of the models met the required assumptions of homoscedasticity of the models' residuals. This was due to the fact that we had too many zeros in the data. Thus, we fitted GLMMs [62] with a binomial error structure and log link function separately for each behaviour. For each GLMM, the number of observations was 432 of 12 dyads. For each model, our response variable was whether the respective variable occurred in a given session (Yes, No) and our only predictor was the variable species (Bonobo, Chimpanzee). To control for any effects that mere time passing might have had on the expression of the respective behaviour, we included the z-transformed variable session number. To control for the fact that the two test conditions most likely induced a different response than the control condition, we included the factor condition with three levels ('Same', 'Different', 'Control'). Finally, to control for any influences that the age of the receiver and helper together might have had on the expression of any of the behaviours (e.g. younger individuals might have reached more when the helper was old), we again included the interaction term between the z-transformed variables age of the helper and age of the receiver, and the two respective main effects. Again, to keep type 1 error rates at the nominal level of 5%, we included the random intercepts for test day and dyad identity. We included the random slopes components [64,65] within dyad identity for the fixed-effects condition and session number, and the random slopes components [64,65] within test day for the fixed-effects species, the interaction between the helpers' and receivers' age, and their main effects.

Furthermore, we investigated whether any of the behaviours executed by the receiver were more likely to occur before versus after the correct tool was shared. We again only used the data of the bonobos given that the chimpanzees only transferred two tools, and fitted a GLMM [62] with a binomial error structure and logit link function [63]. Our response variable was whether the behaviour occurred in a given session (Yes, No), in which a correct tool was transferred, indicated separately for the periods before and after the correct tool transfer and separately for each behaviour that we considered in this model (i.e. reaching, scratching and staying in close proximity). We did not include the behaviour banging as only one individual showed it. Furthermore, only sessions in which a correct tool was eventually shared were included in this analysis and incorrect tools that were shared first were ignored. The rationale behind this is that the need of receiving a tool was only fulfilled if a correct tool was shared. In case no or an incorrect tool was shared, it would have been sensible for the receiver to continue begging. The number of observations was 696 of four dyads. Our key predictor was the variable 'before versus after the correct tool transfer' (hereafter, 'before versus after') to understand whether acquiring the correct tool indeed influenced the occurrence of receivers' behaviours. Additionally, we included the type of behaviour with three levels (reaching, scratching, close proximity) and its interaction 'before versus after' into the model. This interaction term informs

about whether for any of the behaviours the effect of 'before versus after' is different from any of the other behaviours. To give an example, even though scratching might occur significantly less often after than before a correct tool is shared, reaching might in comparison occur even significantly less often after than before that transfer. If the interaction term is not significant but only the main factor assessing the effect 'before versus after', it means the probability of each behaviour being executed by the receiver is significantly different before and after the transfer but to the same degree. We included an offset term [63] to account for the fact that the duration of the periods before and after the transfer differed from one another and also between trials, as the tool was not shared to the exact same time across sessions. The durations were log-transformed in order to add the offset term. To control for any effects that the difference in test conditions might have had on the receiver, we used conditions with two levels ('Same', 'Different'). We did not use the condition 'Control' because any of the behaviours should not have been executed for the reason of acquiring tools, given that the receiver also had access to tools. Additionally, we included experiments with three levels ('Helping 1', 'Cooperation', 'Helping 2') to control for the fact that probabilities of the behaviours' occurrence differed between the three experiments given that there was an incentive to share tools in the experiment 'Cooperation'. Moreover, we included the $z$-transformed variable session number to control for the possibility that the probability of the receivers showing certain behaviours might have in- or decreased over time. As in the previous model, we included the interaction between the $z$-transformed variables age of the helper and age of the receiver, and the two respective main effects. Finally, to keep type 1 error rates at the nominal level of 5%, we included the random intercepts [64,65] for dyad identity, test day identity and trial identity. Moreover, we included the random slopes components [64,65] within dyad identity for the factors behaviour, 'before versus after', experiment, condition, session number, age of helper, age of receiver, the interaction between behaviour and 'before versus after', and the interaction between age of helper and age of receiver. Second, we included the random slopes components within test day identity for the predictors behaviour, 'before versus after', condition, and the interaction between behaviour and 'before versus after'. Third, we included the random slopes components within trial identity for the predictors behaviour, 'before versus after', and the interaction between behaviour and 'before versus after'.

To then further understand whether the possible effect of the variable 'before versus after' was merely apparent because any tool was shared, we fitted the exact same model but looked at whether the behaviours occurred before versus after an incorrect tool was shared in a given session. Thus, the same key and control predictors and random intercept and slope components were used, while the response variable now considered instances of an incorrect transfer instead of correct transfer in that session.

# 3. Results

## 3.1. Tool transfers

Bonobo helpers shared a tool in 142 sessions across all conditions and experiments, and more frequently in both test conditions than the control (figure 4$a$). Bonobos shared most tools in experiment 'Cooperation', in which helpers were given an incentive. Oftentimes, helpers transferred more than one tool in a given session. The total number of tool transfers was 246, with 122 occurring in the condition 'Same', 115 in the condition 'Different' and 9 in the control condition across all three experiments. Four of the six tested bonobo dyads contributed to the number of observed transfers and in each dyad, we found a similar pattern of mainly sharing in the two test conditions (figure 4$b$). These four pairs were composed solely of females, one being a mother–infant dyad. We observed no transfers in the two mixed-sex bonobo pairs.

The transfer frequency of bonobo helpers stands in sharp contrast to that of the chimpanzees. We only observed two tool transfers by a chimpanzee mother to her subadult daughter. Both transfers occurred in the condition 'Same' in the experiment 'Helping' and took place in two separate sessions (one transfer per session). Since none of the chimpanzee helpers shared a tool in the experiment 'Cooperation', we did not switch back to experiment 'Helping 2'.

The GLMM that we used to assess which factors influenced tool transfers was significant ($\chi^2 = 66.76$, d.f. = 8, $N = 576$, $p < 0.001$; see electronic supplementary material, for table). Bonobo helpers shared a tool significantly more often in the two test than the control condition and, hence, were sensitive to whether the receiver was actually in need of a tool ($\chi^2 = 26.93$, d.f. = 2, $p < 0.001$). We also found a significant

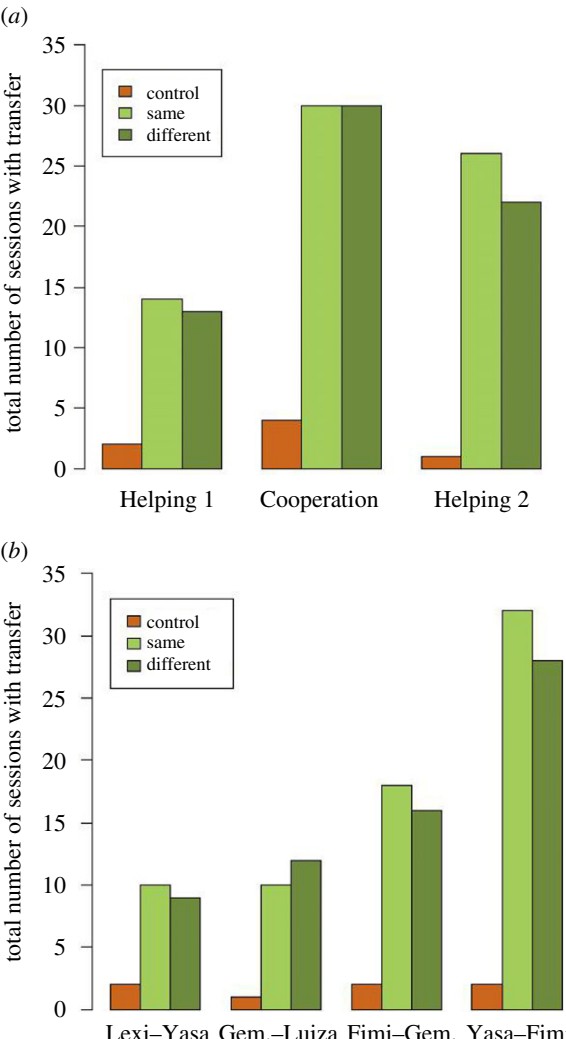

**Figure 4.** Number of sessions with at least one tool transfer in each of the three conditions for each experiment (*a*) and for each dyad across all experiments (*b*). Across all three experiments, every dyad received a total of 36 sessions in each condition. Pair 'Yasa–Fimi' is a mother–infant dyad.

interaction between the type of experiment presented to the dyads and the corresponding session number ($\chi^2 = 13.70$, d.f. = 2, $p = 0.001$). The interaction indicates that the effect of session number on the probability of a transfer occurring differed depending on the type of experiment. Transfers were done randomly across all sessions when helpers did not benefit from sharing a tool. However, when helpers did benefit the probability of a transfer increased over the course of the experiment (figure 5). Reaching by the receiver significantly and positively influenced the probability that a helper transferred a tool in a given session ($\chi^2 = 5.77$, d.f. = 1, $p = 0.016$).

## 3.2. Tool selection

Further, we assessed whether helpers were able to tailor their prosocial acts according to the needs of the receiver. The predictors of the GLMM did not significantly contribute to explaining whether the first transfer in a given session, in which a transfer occurred, was correct ($\chi^2 = 3.40$, d.f. = 4, $N = 135$, $p = 0.49$; see electronic supplementary material, for table). Thus, even though helpers benefited from transferring the correct tool in the experiment 'Cooperation', they did not pay more attention to the needs of the receiver than when they did not benefit from such transfers. It also did not make an obvious difference if helpers needed to share the same tool as the one they had to use themselves (condition 'Same') or a different tool than the own (condition 'Different'). Helpers also did not prefer to share one specific type of tool.

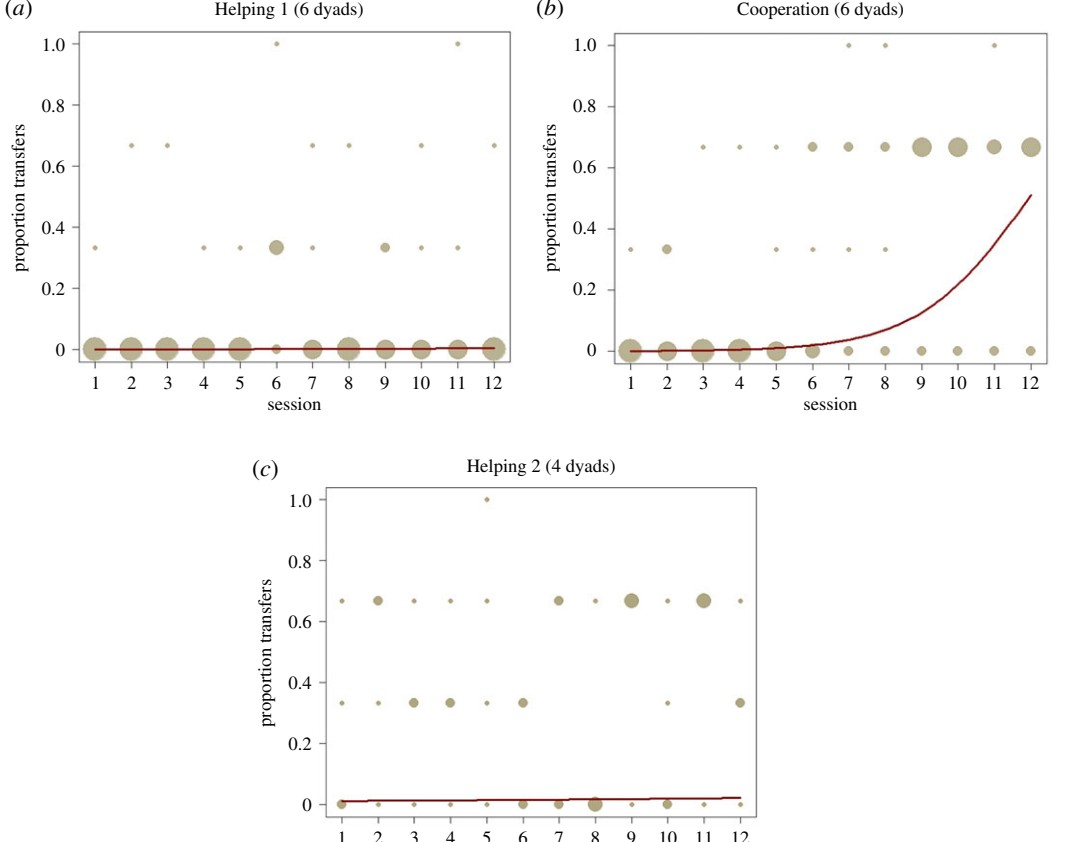

**Figure 5.** Proportion of transfers across the three conditions within each session separately plotted for the experiment 'Helping 1' (a), 'Cooperation' (b) and 'Helping 2' (c). Proportions being 0 denote helpers that did not share a tool in any of the three conditions. Proportions being 0.33 denote helpers that only shared in one condition on that session. Similarly, proportions being 0.66 denote helpers that shared a tool in two conditions on that session. Finally, proportions being 1 denote helpers that shared a tool on all three conditions on that session. The larger the area of the points, the more helpers acted in such a manner on the same session. The interaction of session and experiment can be visually assessed as transfers were done randomly across sessions in experiment 'Helping 1' and 'Helping 2' but increase to a consistent level around session 6 in the experiment 'Cooperation'.

Since the predictors of this model did not significantly contribute to explain when a correct tool was transferred first, we ran two separate models. In the first, the condition 'Same' was the reference group and in the second, condition 'Different' was the reference group while keeping all other predictors centred. Testing the intercept of the first model against a chance level of 2/3 showed a non-significant difference from chance, $p = 0.096$. Testing the intercept of the second model against a chance level of 1/3 also showed a non-significant difference from chance, $p = 0.315$. Therefore, it did not matter if receivers needed the same or a different tool than the helper, in both scenarios, the helper seemingly transferred tools randomly.

## 3.3. Receivers' behaviours

Finally, we analysed whether (i) receivers of the two species behaved differently to one another and (ii) bonobo receivers changed their behaviour upon acquiring the correct tool.

Chimpanzee receivers scratched themselves at least once in more sessions than bonobo receivers ($\chi^2 = 7.47$, d.f. = 1, $N = 432$, $p = 0.006$; see electronic supplementary material, for table), but did not significantly differ to bonobos in any of the other behaviours that we coded. The GLMM that assessed whether receivers of the two species were more likely to reach at least once during a session was not significant ($\chi^2 = 2.67$, d.f. = 1, $N = 432$, $p = 0.102$; see electronic supplementary material, for table). We also did not find a difference regarding close proximity ($\chi^2 = 0.01$, d.f. = 1, $N = 432$, $p = 0.905$; see electronic supplementary material, for table) and banging ($\chi^2 = 1.77$, d.f. = 1, $N = 432$, $p = 0.184$;

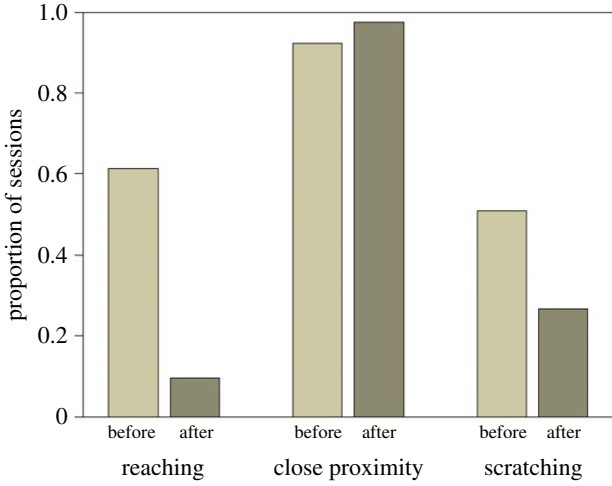

**Figure 6.** Proportion of sessions in which the receivers put their fingers through the mesh (reaching), remained close to the mesh dividing the two cages (close proximity) and scratched themselves (scratching) before and after the correct tool was shared while controlling for the influence of the other predictors in the model.

see electronic supplementary material, for table). The standard errors of the model that investigated the effect of banging were very large due to the general infrequency of banging by bonobo receivers. Thus, the model result is somewhat uncertain. In general, however, behavioural differences seem insufficient to explain the lack of transfers from chimpanzee helpers.

Additionally, we assessed whether the behaviour of bonobo receivers differed before and after they received the correct tool and obtained a significant result ($\chi^2 = 22.31$, d.f. = 3, $N = 696$, $p < 0.001$; see electronic supplementary material, for table). The interaction between the effect of before versus after the correct transfer and the type of behaviour executed by the receiver was highly significant ($\chi^2 = 16.68$, d.f. = 2, $p < 0.001$). This indicates that the degree of change in the likelihood of the receiver executing the behaviour before versus after the correct transfer was significantly different between at least two of the behaviour types. Bonobos stayed in close proximity to a similar degree before and after they received the correct tool (figure 6).

However, they scratched themselves less after the correct transfer occurred. The largest difference concerns reaching: receivers exhibited substantially less reaching after they obtained the correct tool. Bonobo receivers reached in 60% of the sessions before obtaining the necessary tool but they continued reaching only in 10% of the sessions afterwards.

We fitted an additional model, which investigated whether the behaviours occurred before versus after the first incorrect tool was shared in a session. However, the GLMM did not significantly contribute to explaining why the receivers executed the coded behaviours in case an incorrect tool was shared ($\chi^2 = 6.75$, d.f. = 3, $N = 552$, $p = 0.08$; see electronic supplementary material, for table). We thus conclude that the effect of 'before and after' was contingent on whether the correct tool was shared and not just any tool.

## 4. Discussion

While we found multiple spontaneous tool transfers in all female–female bonobo dyads even when they did not directly benefit from it, only two transfers occurred in a mother–daughter chimpanzee dyad and none in the other chimpanzee dyads. Once bonobo helpers benefited from tool transfers and understood the task, they shared tools consistently. By contrast, none of the chimpanzee helpers shared a tool in the cooperative task even though they would have gained rewards by doing so. To control for the influence of gaining more experience over time, we switched back to the helping task for all dyads that shared tools in the cooperative task. While bonobos continued to share tools at a high rate, they did so randomly dispersed across sessions instead of consistently as in the cooperative task. Across experiments, transfers occurred when there was a need for it and not just because another individual was present. Begging influenced the likelihood that tools were shared by bonobo helpers. Even though receivers of both species begged to a similar degree, only bonobo helpers responded by sharing tools to such requests. Bonobo receivers exhibited begging significantly less often once they acquired the correct

tool. We did not find evidence that bonobo helpers tailored the tool selection to the need of the receiver but randomly transferred tools until the correct one was transferred.

Only one of the six chimpanzee helpers transferred a single tool to her subadult daughter in two sessions of the helping task and none of the six helpers transferred any tool in the cooperative task. There might be two possible explanations for this result, one for each experiment. When contrasting our findings from the *helping task* with previous research, which did observe that chimpanzees transfer objects (e.g. [22,23]), one difference is that in such studies, helpers did not receive any other task than transferring objects. In our study, helpers could operate their own apparatus and acquire rewards themselves. They were, therefore, not confronted with the alternative of either transferring an object or doing nothing. As was discussed, Tennie *et al.* [25] found that chimpanzees released a peg that was holding a baited apparatus out of reach from the conspecific at the same rates regardless of whether the result helped or hindered the partner's access to the food. This calls into question why chimpanzees performed the behaviours observed in helping studies. In both their and the original study [20], chimpanzees received a towel soaked in juice and a non-functional rope as distractor items to decrease helpers' manipulation rates and prevent ceiling effects. Melis *et al.* [20] reported a lower response rate (50–55%) in their study compared to previous studies without distractor items (75%, [19]; 80%, [21]). In our study, subjects could retrieve rewards by operating their own apparatus, which was not only a distractor item, but also a goal-directed task. This might have further decreased response rates that were not prosocially motivated. To test this explanation, controlled experiments are needed that not only vary the outcome of the partner but also the degree of involvement in distractor tasks given to the subject.

In the *cooperative task*, chimpanzees would have benefited from transferring a tool, but we still did not observe a single transfer. Given that in wild chimpanzee communities, males seem to cooperate with one another most frequently, we expected to find higher rates of cooperation between males compared to our mixed-sex and female–female dyads. Yet, we did not observe transfers in either of the dyads. It is possible that helpers did not share tools because they might have perceived the task as competitive rather than cooperative. All individuals only completed a non-social training, in which they could access both sides of the apparatuses and subsequently eat all grapes. In the test, chimpanzees might have misperceived the task and expected that all grapes would be delivered to the fastest individual. This could have prevented them from recognizing that the partner did not have a tool. Nevertheless, during the control condition, helpers had the chance to perceive the contingencies of the apparatus and that the rewards were evenly distributed. Yet, apes received the control condition only on one out of three trials per testing day, which might not have been enough to learn the contingencies. To assess and rule out the possibility that chimpanzees perceived the task as competitive, future studies could incorporate a social training for half the sample and then compare both groups.

In comparison to the negligible frequency of transfers done by chimpanzee helpers, we found that female bonobos shared tools with their female partners even when they did not directly benefit. The male bonobo did not transfer any tool to his female partner in either experiment, nor did the female helper share a tool with him. Even though this result is intriguing and might further point to the notion that mainly female bonobos cooperate and support each other, our sample size is too small to draw definite conclusions as the effect might be influenced by this particular male. Large-scale studies are needed to assess if our result holds true.

When bonobo helpers did not benefit from tool transfers, the probability of a transfer did not increase over the course of the experiment and transfers were randomly dispersed over the 12 sessions. When they did benefit in the cooperative task, they transferred tools in every session once they learned how the task works, revealing motivational differences between the two experiments. Nevertheless, bonobos continued to share tools even when we switched back to experiment 'Helping' and they again did not benefit from their help.

Bonobo helpers were sensitive to whether a tool was needed by the receiver and did not transfer tools merely because a partner was present. Still, they did not share the correct tool first. Even though each ape passed the training and only used the correct tools for the respective apparatus, the test situation might have increased their stress levels, which could in turn have reduced their ability to quickly perceive which apparatus was presented to the receiver, and thus which tool was needed. This might have been especially true for the more complex apparatuses of the cooperation task. Alternatively, the number of tools could have reduced the pressure to keep track of which tool was needed by the partner. Transferring several tools in a session came at no direct cost as (i) the tools were still useful after using it on the own apparatus and (ii) we provided a spare tool of the type that the helper needed to use. A follow-up experiment with less complex apparatuses, and in which the helpers are forced to decide between both tool types, instead of having access to all, could be used to answer whether bonobos are able to distinguish between their own and others' needs.

Even though bonobos could have also shared the food they acquired themselves in the helping task, they never did so and only transferred tools. Previous studies only observed bonobos to transfer food instead of objects [66,67]. This is the first study to show that bonobos also transfer objects in order to help a conspecific. In comparison to Krupenye *et al.* [67], bonobos in our study could retrieve food that is easily consumed without any prior processing. Nevertheless, it would have been possible for the helpers to respond to their partner's begging by transferring grapes as well. We think the most likely reason why bonobos transferred tools but not food in our study is that in this way they were able to maximize benefits for themselves and their partner. If they had only transferred grapes, they would have lost their food while their partner's food remained inaccessible. Only by retrieving and consuming their own food and transferring a tool to their partner could both benefit. This is the most salient difference to the study by Krupenye *et al.* [67] and future studies could make an effort to combine the two designs in order to clarify possible explanations.

Bonobo receivers reached less often after they received the correct tool and this drop was contingent on whether the correct tool was shared and not just any tool. Even though the individual was occupied with retrieving the food and consuming it upon receiving the correct tool, in most cases, there was still enough time left to continue begging. We also wanted to rule out the possibility that receivers were just reaching to acquire any tool and did not specifically request the correct one. Corroborating existing literature (e.g. [20–22]), reaching of receivers significantly influenced the likelihood that bonobo helpers shared a tool. This finding did not hold true for chimpanzees, given that we did not find species differences in reaching. However, larger samples are needed to draw more solid conclusions regarding a general species difference in terms of the ability or motivation to help conspecifics in need. The sharing-under-pressure hypothesis suggests that food or object sharing in apes can be explained by requests and resulting harassment of the partner and not due to a prosocial motivation [24]. Even though we think that the impact of sticking fingers through the mesh should be minimal in terms of harassment as seen in feeding contexts, this could explain the occurrence of transfers by bonobo helpers but not the lack of transfers by chimpanzee helpers.

One explanation might be that the two species differ in how susceptible they are to such harassment with chimpanzees possibly needing cues that are more salient. In previous studies [20,21], chimpanzees could either insert their entire arm into the helper's cage or rattle at a metal chain to draw attention to themselves. Chimpanzees could therefore request much more saliently compared to our study where they could only stick their fingers through the mesh. Bonobos have not been tested with such paradigms yet, so it is unclear whether and how such more explicit forms of requesting would influence their tendency to help. An alternative explanation might be that bonobos are better able to interpret such behaviours, possibly due to enhanced empathetic abilities. As discussed above, bonobos seem more adept at processing socially relevant stimuli [34,37–39] and could have benefited from their possibly heightened awareness of their partners' needs in our study. Either of these explanations might have helped bonobos to understand the cooperative task more easily than chimpanzees. We found lower rates of transfers in the first than in the second helping task, which could suggest that bonobo helpers learned to better interpret the reaching behaviour of their conspecifics during the cooperative task. Through this, they were either able to act more prosocially or were more susceptible to harassment in the second helping task than in the first. Our results are not in line with a mere carry-over effect from the cooperation to the second helping task. In such a case, we would expect higher rates of transfers at the beginning with a subsequent decrease once the individuals recognize that they can retrieve the rewards by themselves. Instead and similar to the first helping task, we find a random distribution of transfers across the 12 sessions of the second helping task.

While we did include a balanced sex composition of chimpanzee dyads (i.e. two pairs of m–m, f–f and m–f), due to logistical reasons that was not possible for bonobos (i.e. four f–f and two m–f pairs). Research from the wild showed that chimpanzees support each other and share food mainly between males, sometimes from males to females and infrequently between females [8–10]. By contrast, coalition formation and food sharing is highest between female bonobos, rare from females to males, and nearly absent between males [30–32]. We, therefore, expected male–male chimpanzee and female–female bonobo dyads to display the highest levels of cooperation in our task, with some possible cooperation or helping among different-sex pairs. However, we only observed cooperation and helping among female bonobo dyads. Even though male wild chimpanzees behave prosocially towards each other in various ways, and the males included in this study showed high tolerance levels and were judged to have a positive relationship with each other, they did not cooperate in our experiment. While several studies have documented a positive influence of relationship quality on cooperative success (e.g. [5,68–70]), others have reported a negligible influence (e.g. [71–73]). It is,

therefore, unclear to what extent relationship quality can proximately explain cooperation or whether it is even sufficient to produce such behaviour [73,74].

Even though we lack quantitative data on the relationship quality and tolerance between individuals, there are three reasons that lead us to believe that this factor alone was not responsible for the lack of cooperation between chimpanzees. First, we paired chimpanzees that were judged by the caretakers to have a good relationship based on regular grooming and spatial closeness. Two brothers, in particular, regularly supported each other during conflicts and were identified as having a strongly positive relationship, and yet no cooperation between them was observed in our task. Second, spatial separation, which invariably occurred in our set-up, reduces the influence of relationship quality in various species [7,54,73]. Third, even in studies that reported an influence of relationship quality, cooperation was not entirely absent for those dyads with less affiliative relationships (e.g. [5,69,70,73]). Nevertheless, quantifying the affiliative relationship and comparing free to forced partner choice paradigms will be helpful to solve the question of whether affiliation influences cooperation or whether it is a by-product of tolerance [73].

Interestingly, wild bonobo populations, in comparison to chimpanzee populations, do not show collective behaviours such as border patrolling, hunting or aggressive territorial defence [27–29]. Moreover, due to a high abundance of food in bonobo habitats, there is less need to form coalitions with other group members that might be useful during feeding competition [33]. Therefore, from these observations, we would expect that chimpanzees would be better than bonobos at cooperating strategically with one another. The chimpanzees in our study were not able to either perceive the task as collaborative or did not act accordingly, while most of the bonobos did. We tested captive chimpanzees and bonobos, which reduces the generalizability of our findings to wild populations. It is yet unknown to what extent captive settings influence the cognitive ability and behavioural responses of chimpanzees and bonobos, and if the two species are influenced similarly. One study suggests that chimpanzees might show more abnormal behaviours than bonobos in captive settings; however, as the authors advise additional studies need to directly compare the two species [75]. Moreover, subjects rearing history needs to be considered in such studies. Pomerantz *et al.* [76] suggest that factors such as home range size and party size might be relevant predictors to understand differences in welfare in captive primate species, with wide-ranging species that live in large groups faring worse than those with a more restricted movement patterns and smaller groups. Some home ranges of savannah living chimpanzees have been documented to be as large as 560 km$^2$ [77], while the largest ones of bonobos did not exceed a range of 32 km$^2$ [78]. Nevertheless, most home ranges of chimpanzees are similar to those of bonobos, especially in forest living chimpanzees [79]. Relative party size, on the other hand, tends to be larger in bonobos than chimpanzees [80], but also shows great variation within the two species [80]. Thus, deriving welfare predictions of captive bonobos and chimpanzees from their natural habitat seems to pose difficulties and large-scale assessments of zoo-living populations would be needed. Platforms such as ZIMS (Zoological Information Management System) could aid this endeavour and answer interesting questions to the adaptability and cognitive flexibility of both species. It might also be a chance to understand which behaviours are most relevant to both species and thus induce abnormal behaviours in captivity.

In conclusion, we found that bonobos were more likely to help and cooperate with a conspecific than chimpanzees. Bonobos transferred tools irrespective of whether they gained an immediate benefit from it. Even though they only transferred tools when needed, bonobos did not tailor their help to their partner's actual requirements, contrasting evidence of chimpanzees' targeted helping abilities [22]. Given our small sample size, caution is required with regard to the generalizability of our findings and future studies are needed to assess group differences across both species. The results of this study underline the fact that we need to directly compare bonobos and chimpanzees and use the same design for both species if we want to understand which factors influence the expression of prosocial behaviours and to better understand the evolution of human cooperation.

Ethics. This study was approved both by the ethics committee of the University St Andrews, Scotland, and Max Planck Institute for Evolutionary Anthropology in Germany. Research was non-invasive and strictly adhered to the German legal requirements. This study and the housing conditions of the apes complied with the ethical guidelines of the European and World Association of Zoos and Aquariums (EAZA and WAZA).
Data accessibility. Data available from the Dryad Digital Repository: https://doi.org/10.5061/dryad.1jwstqjsp [81].
Authors' contributions. S.N. and J.C. conceived the study. S.N. conducted the research, coded and analysed the data. S.N. and J.C. wrote and revised the paper.
Competing interests. We declare we have no competing interests.

Funding. This research has received funding from the European Research Council (ERC) under the European Union's Seventh Framework Programme grant no. (FP7/2017-2013) under grant agreement no. 609819-SOMICS.

Acknowledgements. We thank Jorg Massen and two anonymous reviewers for their comments which helped us improve this manuscript. Further, we thank Raik Piesek and Johannes Grossmann for building the apparatuses, Hanna Petschauer for providing reliability coding, Sylvio Tükpe for the set-up figures, Colleen Stephens and Roger Mundry for their statistical support, and the caretakers at WKPRC for enabling testing. Above all, we thank the chimpanzees and bonobos for their participation.

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
