## [Peer Review File · Royal Society Open Science]

Review History

RSOS-191273.R0 (Original submission)

Review form: Reviewer 1

Is the manuscript scientifically sound in its present form?

No

Are the interpretations and conclusions justified by the results?

No

Is the language acceptable?

Yes

Do you have any ethical concerns with this paper?

Yes

Have you any concerns about statistical analyses in this paper?

Yes

Recommendation?

Reject

Comments to the Author(s)

See attached file (Appendix A).

Review form: Reviewer 2 (Jorg Massen)**Is the manuscript scientifically sound in its present form?**

Yes

Are the interpretations and conclusions justified by the results?

Yes

Is the language acceptable?

Yes

Do you have any ethical concerns with this paper?

No

Have you any concerns about statistical analyses in this paper?

No

Recommendation?

Accept with minor revision (please list in comments)

Comments to the Author(s)

Dear Authors,

I was already very positive about your study in the previous review round and think that you also did a nice job with regard to the revision. I have only a few minor comments left, yet do advise publication.

First, and this is the largest point I have, I still believe that the paper should be imbedded in a broader literature and not just in the literature about pan. Especially when looking at the evolution of cooperation, it may be worthwhile to also discuss results on other species (see for example the February issue of this year of *Ethology* and its editorial: Massen 2020 *ethology*). I agree that the discussion about differences between dogs and wolves might fall outside the scope of this paper as there the additional confound of artificial selection comes into play. I do think, however, that there are other species to consider. I made some suggestions below.

Second, as far as I know there are no restrictions to the amount of figures one can have in *RSOS* and then I think the figures depicting the different apparatuses should be in the main text, as it makes it much easier for the reader to grasp.

l. 44 Animal instead of primate

l. 67 See also the recent results positive on African Grey parrots of Brucks & von Bayern (2020) *Curr. Biol.*, which use an exact copy of a paradigm used by Massen et al. 2015 *Frontiers in Comp Psychol*, which failed to show any prosocial motivation in ravens

l.84 see also Schawb et al. 2012 *PLoS One*

- l. 294 rephrase, now the subject of the sentence changes halfway.
- l. 365. as asked before. Which data was used for subsequent analyses? that of the original coder or that of the second coder in those cases where they did not agree?
- l. 604 Maybe add a figure 6b in which you show the same results but now for when the incorrect tool was transferred as to truly contrast them against each other.
- l. 614 While we found MULTIPLE spontaneous...(I think this would stress the difference even more between the bonobos (a lot of transfers) and the chimps (only two)

It was a pleasure reading this interesting study and I look forward seeing it published.

Kind regards,
Jorg Massen

Review form: Reviewer 3

Is the manuscript scientifically sound in its present form?

Yes

Are the interpretations and conclusions justified by the results?

No

Is the language acceptable?

Yes

Do you have any ethical concerns with this paper?

No

Have you any concerns about statistical analyses in this paper?

No

Recommendation?

Major revision is needed (please make suggestions in comments)

Comments to the Author(s)

I read the manuscript and also the correspondence between the authors and the reviewers in the previous review round. I acknowledge that the empirical data on chimpanzees' and bonobos' helping and cooperation is very important and worth publication in the current situation with still little prior research; however, I also agree with the reviewers' concerns.

As the reviewers pointed out, the biggest limitation of this study is the small sample size and the imbalance of the participants' sex ratio between the chimpanzees and bonobos. I understand this often takes place in empirical studies with great apes, but this actually makes it very difficult to generalize the results and discuss the chimpanzee-bonobo species differences. One of the simplest interpretation of the results may be that in mixed-sex pairs, both in chimpanzees and bonobos, individuals are reluctant to hand tool to their partner of different sex. This is irrelevant to the species difference. Thus the authors should tone down their claim of species difference. They should wait for further empirical studies with larger samples or meta-analyses.

Instead, I suggest that the authors focus more specifically on the bonobos' data with more detailed analyses. This report should be important since there are still few studies on bonobos' helping. And it seems interesting that the bonobos are helpful but their helping is not corresponding to their partner's need, which can be discussed in contrast with the previous chimpanzees' targeted helping.

I also wonder how the authors decided the pairing of the individuals. Existing social relationships should have influenced the results. To avoid this arbitrary pairing (as well as to increase the sample size), there should have been a possibility to make all the possible pairings in theory (6*5 pairs respectively in chimpanzees and bonobos). Why didn't the authors design like this?

The other difficult point to interpret is that the chimpanzees practically never demonstrated helping and cooperation. This contradicts the results of the previous studies, and the authors' explanation is not so appealing. There should be many other possible factors that may have affected the results, such as the participants' experience in previous experiments (especially in helping tasks, if they have participated), sex, social relationship between the paired individuals, interaction, explicitness of request behaviors etc. With these results, it is too early to conclude that chimpanzees are reluctant to help others. This should be context-dependent. This also makes the species comparison difficult.

As the present and previous studies suggested, explicit demonstration of request behavior should be important for the apes' helping. Is there any index about the explicitness of request behaviors? Can the authors discuss differences of explicitness in request behaviors between the chimpanzees and bonobos, and also between the present and previous studies? Please also report the ratios of transfer types such as proactive and reactive transfers.

I did not find any captions for the figures.

Decision letter (RSOS-191273.R0)

27-Feb-2020

Dear Ms Nolte:

Manuscript ID RSOS-191273 entitled "Targeted Helping and Cooperation in Zoo-living Chimpanzees and Bonobos" which you submitted to Royal Society Open Science, has been reviewed. The comments from reviewers are included at the bottom of this letter.

In view of the criticisms of the reviewers, the manuscript has been rejected in its current form. However, a new manuscript may be submitted which takes into consideration these comments.

Please note that resubmitting your manuscript does not guarantee eventual acceptance, and that your resubmission will be subject to peer review before a decision is made.

Once you have revised your manuscript, go to <https://mc.manuscriptcentral.com/rsos> and login to your Author Center. Click on "Manuscripts with Decisions," and then click on "Create a

Resubmission" located next to the manuscript number. Then, follow the steps for resubmitting your manuscript.

Your resubmitted manuscript should be submitted by 26-Aug-2020. If you are unable to submit by this date please contact the Editorial Office.

on behalf of Dr Atsushi Iriki (Associate Editor) and Kevin Padian (Subject Editor)
openscience@royalsociety.org

Subject Editor Comments to Author (Professor Kevin Padian):

Comments to the Author:

Dear authors, thank you for your revisions. We have a range of opinions but some very strong concerns about the study and also about your responsiveness to previous concerns. Small sample size, biased sex ratio, interpretation of behavior of bonobos and chimps in zoos, and interpretation of results are all expressed as problems. If you choose to resubmit, please be sure to address all issues thoroughly; we will not be able to send it back to you for further revision. Thanks for considering RSOS.

Reviewers' Comments to Author:

Reviewer: 1

Comments to the Author(s)

See attached file ("Review of Nolte and Call.pdf")

Reviewer: 2

Comments to the Author(s)

Dear Authors,

I was already very positive about your study in the previous review round and think that you also did a nice job with regard to the revision. I have only a few minor comments left, yet do advise publication.

First, and this is the largest point I have, I still believe that the paper should be imbedded in a broader literature and not just in the literature about pan. Especially when looking at the evolution of cooperation, it may be worthwhile to also discuss results on other species (see for example the February issue of this year of *Ethology* and its editorial: Massen 2020 *ethology*). I agree that the discussion about differences between dogs and wolves might fall outside the scope of this paper as there the additional confound of artificial selection comes into play. I do think, however, that there are other species to consider. I made some suggestions below.

Second, as far as I know there are no restrictions to the amount of figures one can have in RSOS

and then I think the figures depicting the different apparatuses should be in the main text, as it makes it much easier for the reader to grasp.

l. 44 Animal instead of primate

l. 67 See also the recent results positive on African Grey parrots of Brucks & von Bayern (2020) *Curr. Biol.*, which use an exact copy of a paradigm used by Massen et al. 2015 *Frontiers in Comp Psychol*, which failed to show any prosocial motivation in ravens

l.84 see also Schawb et al. 2012 *PLoS One*

l. 294 rephrase, now the subject of the sentence changes halfway.

l. 365. as asked before. Which data was used for subsequent analyses? that of the original coder or that of the second coder in those cases where they did not agree?

l. 604 Maybe add a figure 6b in which you show the same results but now for when the incorrect tool was transferred as to truly contrast them against each other.

l. 614 While we found MULTIPLE spontaneous....(I think this would stress the difference even more between the bonobos (a lot of transfers) and the chimps (only two)

It was a pleasure reading this interesting study and I look forward seeing it published.

Kind regards,
Jorg Massen

Reviewer: 3

Comments to the Author(s)

I read the manuscript and also the correspondence between the authors and the reviewers in the previous review round. I acknowledge that the empirical data on chimpanzees' and bonobos' helping and cooperation is very important and worth publication in the current situation with still little prior research; however, I also agree with the reviewers' concerns.

As the reviewers pointed out, the biggest limitation of this study is the small sample size and the imbalance of the participants' sex ratio between the chimpanzees and bonobos. I understand this often takes place in empirical studies with great apes, but this actually makes it very difficult to generalize the results and discuss the chimpanzee-bonobo species differences. One of the simplest interpretation of the results may be that in mixed-sex pairs, both in chimpanzees and bonobos, individuals are reluctant to hand tool to their partner of different sex. This is irrelevant to the species difference. Thus the authors should tone down their claim of species difference. They should wait for further empirical studies with larger samples or meta-analyses.

Instead, I suggest that the authors focus more specifically on the bonobos' data with more detailed analyses. This report should be important since there are still few studies on bonobos' helping. And it seems interesting that the bonobos are helpful but their helping is not corresponding to their partner's need, which can be discussed in contrast with the previous chimpanzees' targeted helping.

I also wonder how the authors decided the pairing of the individuals. Existing social relationships should have influenced the results. To avoid this arbitrary pairing (as well as to increase the sample size), there should have been a possibility to make all the possible pairings in theory (6*5 pairs respectively in chimpanzees and bonobos). Why didn't the authors design like this?

The other difficult point to interpret is that the chimpanzees practically never demonstrated helping and cooperation. This contradicts the results of the previous studies, and the authors' explanation is not so appealing. There should be many other possible factors that may have affected the results, such as the participants' experience in previous experiments (especially in helping tasks, if they have participated), sex, social relationship between the paired individuals, interaction, explicitness of request behaviors etc. With these results, it is too early to conclude that chimpanzees are reluctant to help others. This should be context-dependent. This also makes the species comparison difficult.

As the present and previous studies suggested, explicit demonstration of request behavior should be important for the apes' helping. Is there any index about the explicitness of request behaviors? Can the authors discuss differences of explicitness in request behaviors between the chimpanzees and bonobos, and also between the present and previous studies? Please also report the ratios of transfer types such as proactive and reactive transfers.

I did not find any captions for the figures.

Author's Response to Decision Letter for (RSOS-191273.R0)

See Appendices B - D.

RSOS-201688.R0

Review form: Reviewer 1

Is the manuscript scientifically sound in its present form?

Yes

Are the interpretations and conclusions justified by the results?

Yes

Is the language acceptable?

Yes

Do you have any ethical concerns with this paper?

Yes

Have you any concerns about statistical analyses in this paper?

Yes

Recommendation?

Accept with minor revision (please list in comments)

Comments to the Author(s)

I thank the authors for considering my comments.

Review form: Reviewer 3

Is the manuscript scientifically sound in its present form?

Yes

Are the interpretations and conclusions justified by the results?

Yes

Is the language acceptable?

Yes

Do you have any ethical concerns with this paper?

No

Have you any concerns about statistical analyses in this paper?

No

Recommendation?

Accept with minor revision (please list in comments)

Comments to the Author(s)

I read the authors' replies to my previous review comments, and found that they are clear except one point. Though my concerns about the small sample size and the selection of pairs still remain, I also understand the restriction.

I would like to raise a few more comments which may hopefully be helpful to improve the manuscript.

1. The authors say "The social relationship could influence the transfer frequency in the helping task, however, it should not lead to zero transfers in the cooperation task where helpers would gain immediate benefits from tool transfers." In their reply to my review comment, but I don't agree with this. Even in humans, people do not cooperate with a partner in bad relationship even when it is mutually beneficial contexts (see politics everywhere in the world...). In the Methods section, the authors should mention more clearly how they selected the pairs from the larger pool of possible candidates (how many chimpanzees and bonobos were there in MPI?) and why they didn't test the others.

2. When discussing cooperation, especially prosociality, among wild chimpanzees and bonobos, it is more relevant to discuss food sharing, a typical dyadic prosocial behavior in the wild. There have been several studies both in chimpanzees and bonobos, which would enhance the discussion on species comparison (e.g. males share food with others in chimpanzees while females share food with others in bonobos).

3. There should be some other interpretation on the result that the bonobos did not select appropriate tools according to the partner's need. One plausible explanation might be that the apparatus was complicated and not so intuitive, and it might have been difficult for a helper to see which tool is needed on the partner's side. Though they could "understand" how the apparatus works, this ability may be exercised for his/her own reward, but not for the others when it requires some cognitive load.

4. The chimpanzees and the bonobos were tested in the same experimental rooms, weren't they? The exact size of the rooms is informative.

Decision letter (RSOS-201688.R0)

Dear Ms Nolte

The Editors assigned to your paper RSOS-201688 "Targeted Helping and Cooperation in Zoo-living Chimpanzees and Bonobos" have now received comments from reviewers and would like you to revise the paper in accordance with the reviewer comments and any comments from the Editors. Please note this decision does not guarantee eventual acceptance.

Please submit your revised manuscript and required files (see below) no later than 21 days from today's (ie 26-Oct-2020) date. Note: the ScholarOne system will 'lock' if submission of the revision is attempted 21 or more days after the deadline. If you do not think you will be able to meet this deadline please contact the editorial office immediately.

on behalf of Dr Atsushi Iriki (Associate Editor) and Kevin Padian (Subject Editor)
openscience@royalsociety.org

Editor comments:

Thank you for your efforts at revising and resubmitting. One reviewer is happy at this point but the other continues to have serious concerns. These need to be answered very conscientiously, please, in your final revision. Note that we will send the MS to this reviewer for final comment. If

issues still remain we will unfortunately not be able to consider the manuscript further. Thanks again.

Reviewer comments to Author:

Reviewer: 1

Comments to the Author(s)

I thank the authors for considering my comments.

Reviewer: 3

Comments to the Author(s)

I read the authors' replies to my previous review comments, and found that they are clear except one point. Though my concerns about the small sample size and the selection of pairs still remain, I also understand the restriction.

I would like to raise a few more comments which may hopefully be helpful to improve the manuscript.

1. The authors say "The social relationship could influence the transfer frequency in the helping task, however, it should not lead to zero transfers in the cooperation task where helpers would gain immediate benefits from tool transfers." In their reply to my review comment, but I don't agree with this. Even in humans, people do not cooperate with a partner in bad relationship even when it is mutually beneficial contexts (see politics everywhere in the world...). In the Methods section, the authors should mention more clearly how they selected the pairs from the larger pool of possible candidates (how many chimpanzees and bonobos were there in MPI?) and why they didn't test the others.

2. When discussing cooperation, especially prosociality, among wild chimpanzees and bonobos, it is more relevant to discuss food sharing, a typical dyadic prosocial behavior in the wild. There have been several studies both in chimpanzees and bonobos, which would enhance the discussion on species comparison (e.g. males share food with others in chimpanzees while females share food with others in bonobos).

3. There should be some other interpretation on the result that the bonobos did not select appropriate tools according to the partner's need. One plausible explanation might be that the apparatus was complicated and not so intuitive, and it might have been difficult for a helper to see which tool is needed on the partner's side. Though they could "understand" how the apparatus works, this ability may be exercised for his/her own reward, but not for the others when it requires some cognitive load.

4. The chimpanzees and the bonobos were tested in the same experimental rooms, weren't they? The exact size of the rooms is informative.

===PREPARING YOUR MANUSCRIPT===

a 'clean' version of the new manuscript that incorporates the changes made, but does not highlight them. This version will be used for typesetting if your manuscript is accepted. Please ensure that any equations included in the paper are editable text and not embedded images.

===PREPARING YOUR REVISION IN SCHOLARONE===

- If you are requesting a discretionary waiver for the article processing charge, the waiver form must be included at this step.
- If you are providing image files for potential cover images, please upload these at this step, and inform the editorial office you have done so. You must hold the copyright to any image provided.
- A copy of your point-by-point response to referees and Editors. This will expedite the preparation of your proof.

- Ensure that your data access statement meets the requirements at <https://royalsociety.org/journals/authors/author-guidelines/#data>. You should ensure that you cite the dataset in your reference list. If you have deposited data etc in the Dryad repository, please include both the 'For publication' link and 'For review' link at this stage.
- If you are requesting an article processing charge waiver, you must select the relevant waiver option (if requesting a discretionary waiver, the form should have been uploaded at Step 3 'File upload' above).
- If you have uploaded ESM files, please ensure you follow the guidance at <https://royalsociety.org/journals/authors/author-guidelines/#supplementary-material> to include a suitable title and informative caption. An example of appropriate titling and captioning may be found at https://figshare.com/articles/Table_S2_from_Is_there_a_trade-off_between_peak_performance_and_performance_breadth_across_temperatures_for_aerobic_scop_e_in_teleost_fishes_/3843624.

Author's Response to Decision Letter for (RSOS-201688.R0)

See Appendix E.

RSOS-201688.R1 (Revision)

Review form: Reviewer 3

Is the manuscript scientifically sound in its present form?

Yes

Are the interpretations and conclusions justified by the results?

Yes

Is the language acceptable?

Yes

Do you have any ethical concerns with this paper?

No

Have you any concerns about statistical analyses in this paper?

No

Recommendation?

Accept as is

Comments to the Author(s)

The authors acknowledged the problematic points I raised well in the manuscript, and this is acceptable.

Decision letter (RSOS-201688.R1)

Dear Ms Nolte,

It is a pleasure to accept your manuscript entitled "Targeted Helping and Cooperation in Zoo-living Chimpanzees and Bonobos" in its current form for publication in Royal Society Open Science. The comments of the reviewers who reviewed your manuscript are included at the foot of this letter.

Best regards,
Lianne Parkhouse
Editorial Coordinator
Royal Society Open Science

on behalf of Dr Atsushi Iriki (Associate Editor) and Kevin Padian (Subject Editor)
openscience@royalsociety.org

Reviewer comments to Author:

Reviewer: 3

Comments to the Author(s)

The authors acknowledged the problematic points I raised well in the manuscript, and this is acceptable.

Appendix A

Review of Bonobo and Chimpanzee zoo compared by J. Call

When one wants to compare cognitive differences between two species, it is mandatory to know about the difference in the life history and social behavior of the species. Naturally, none of them is found in zoos nor in captive settings, but they are found in the tropical African environment. An essential prerequisite for such a comparison is therefore first to know what are the cooperation and prosocial tendencies of the two species in their natural environment, and second if the two species really react in the same way to captive living conditions. Sadly, none of this is discussed in the present paper, and as long as they cannot show that captive studies are representative for the species, the value of the present study should be more precisely limited.

First point, at last some direct comparisons of social behaviors between bonobos and chimpanzees are available that contradicts the generalizability of the claims of captive bonobos being more cooperative than captive chimpanzees. Such works needs to be read and cited: Surbeck et al. 2017. *American Journal of Primatology*, 79(6): e22641. Surbeck et al. 2017. Sex-specific association patterns in bonobos and chimpanzees reflect species differences in cooperation. *Royal Society Open Science*, 4: 161081. From these studies, it is clear that the authors misrepresent the differences between the two species in nature.

Second point, a large number of studies on animal welfare documents the fact that even closely related species might suffer from captive living conditions in very different ways (e.g. Clubb and Mason 2003, Mason 2010, Pomerantz et al. 2013). This needs to be addressed as conclusions from captive living animals cannot be directly reported on wild animals. Both points are ignored by the authors and need to be addressed. Bonobos and chimpanzees seem to suffer quite differently from captive conditions and these raises fascinating questions about how this is so. This may also explain why so many mix results are coming out from such captive experimental studies.

Finally the reviewer 2 on the previous version commented on the small sample size and bias sex sampled across the two species, as well as the inconclusiveness to add another experiment to the pile. I agree with reviewer 2 and I am surprised the authors simply ignores this basic issue.

Appendix B

Dear reviewer 1

We would like to thank you for your comments and suggestions.

While we did include that 1) wild bonobos in contrast to wild chimpanzees do not show cooperative behaviours such as boundary patrols, collective hunting, or aggressive territorial defense (line 115-118) and 2) that there is no clear evidence that bonobos cooperate more than chimpanzees in experimental settings (line 180-200), we found your suggestions very helpful to enhance the depth of our introduction and discussion.

Information regarding differences between bonobo and chimpanzee association patterns and coalition formation are incorporated into the paper and can be found at:

Line 59-61 “Most of these behaviours such as boundary patrols, coalition formation, and collective hunting occur between male chimpanzees (8-10, 12).”

Line 122-129 “While chimpanzees tend to associate most strongly with same-sex partners, male bonobos associate most strongly with female bonobos (i.e. their mothers) and female bonobos with other females though to a lesser degree than in chimpanzees (34). Surbeck and colleagues (34) argue that such association patterns might reflect the tactical selection of potentially best cooperation partners. It might therefore explain high male-male cooperation levels in chimpanzees and increased mating success for bonobo males with strong associations with their mothers (34). Thus, association patterns within a group might be used as a proxy to potentially infer who the best cooperative partners are.”

The review by Mason is very interesting and informative, thanks. We are aware of the diminished generalizability to wild populations, which is why we specifically underlined the fact that we talk about zoo-living populations in the title. However, we are happy to elaborate on this issue in the paper and also find it an interesting avenue for future research. Please see on line 745-771:

“Interestingly, wild bonobo populations, in comparison to chimpanzee populations, do not show collective behaviours such as border patrolling, hunting, or aggressive territorial defense (28-30). Moreover, due to a high abundance of food in bonobo habitats there is less

need to form coalitions with other group members that might be useful during feeding competition (34). Therefore, from these observations we would expect that chimpanzees would be better than bonobos at cooperating strategically with one another. The chimpanzees in our study were not able to either perceive the task as collaborative, or did not act accordingly, while most of the bonobos did. We tested captive chimpanzees and bonobos, which reduces the generalizability of our findings to wild populations. It is yet unknown to what extent captive settings influence the cognitive ability and behavioural responses of chimpanzees and bonobos, and if the two species are influenced similarly. One study suggests that chimpanzees might show more abnormal behaviours than bonobos in captive settings, however, as the authors advise additional studies need to directly compare the two species (70). Moreover, subjects rearing history needs to be considered in such studies. Pomerantz and colleagues (71) suggest that factors such as home range size and party size might be relevant predictors to understand differences in welfare in captive primate species, with wide-ranging species that live in large groups faring worse than those with a more restricted movement patterns and smaller groups. Some home ranges of savanna living chimpanzees have been documented to be as large as 560km² (72), while the largest ones of bonobos did not exceed a range of 32km² (73). Nevertheless, most home ranges of chimpanzees are similar to those of bonobos, especially in forest living chimpanzees (74). Relative party size on the other hand tends to be larger in bonobos than chimpanzees (75). However, again there is great variation within the two species (75). Thus, deriving welfare predictions of captive bonobos and chimpanzees from their natural habitat seems to pose difficulties and large scale assessments of zoo-living populations would be needed. Platforms such as ZIMS (Zoological Information Management System) could aid this endeavour and answer interesting questions to the adaptability and cognitive flexibility of both species. It might also be a chance to understand which behaviours are most relevant to both species and thus induce abnormal behaviours in captivity.”

We further elaborated on the sex composition and sample size of our study. We therefore adapted the original text as follows:

Original text, line 668-672 in old manuscript: “Even though this result is intriguing and might further point to the notion that mainly female bonobos cooperate and support each other, our sample size is too small to draw definite conclusions as the effect might be influenced by

this particular male. Large scale studies including more mixed-sex and additionally male-male dyads are needed to assess if our result holds true.”

Changed to, line 677-684 in new manuscript: “Even though this result is intriguing and might further point to the notion that mainly female bonobos cooperate and support each other, our sample size is too small to draw definite conclusions as the effect might be influenced by this particular male. While we did include a balanced sex composition of chimpanzee dyads (i.e. two pairs of m-m, f-f, and m-f), due to logistical reasons that was not possible for bonobos. Based on research from the wild (8-10, 12, 30, 33-34), however, we expected male-male chimpanzee and female-female bonobo dyads to display the highest levels of cooperation, but we only observed cooperation and helping in the latter. However, large scale studies are needed to assess if our result holds true.”

Furthermore, we included on line 776-778: “Given our small sample size, caution is required with regard to the generalizability of our findings and future studies are needed to assess group differences across both species.”

Finally, regarding the significance of this work, we think that this study contributes in several ways. It is, to our knowledge, the first to show that bonobos transfer tools to one another (a previous study reported that they did not transfer tools), which has interesting implications considering that this species is often not described as an adept tool user. Even though the complex design (and limited access to large bonobo groups) did not allow for a large sample size, it provides valuable insight into the mechanisms underlying cooperation and altruism. The method is also quite unique in directly pitting cooperation against helping. Given the small sample sizes that are customary in apes studies, it is by the accumulation of such carefully designed studies (ideally that use the same method) that one can hope to be able to draw more generalizable and theories.

Thank you for your constructive feedback and best wishes,

Nolte & Call

Appendix C

Dear Jorg Massen

We would like to thank you for your comments and suggestions. We appreciate your constructive feedback and your motivation to improve the current paper. Please find our answers below your comments.

1. First, and this is the largest point I have, I still believe that the paper should be imbedded in a broader literature and not just in the literature about pan. Especially when looking at the evolution of cooperation, it may be worthwhile to also discuss results on other species (see for example the February issue of this year of *Ethology* and its editorial: Massen 2020 *ethology*). I agree that the discussion about differences between dogs and wolves might fall outside the scope of this paper as there the additional confound of artificial selection comes into play. I do think, however, that there are other species to consider. I made some suggestions below.

→ Please see changes made to the first paragraph of the paper

2. Second, as far as I know there are no restrictions to the amount of figures one can have in RSOS and then I think the figures depicting the different apparatuses should be in the main text, as it makes it much easier for the reader to grasp.

→ We included the pictures of each apparatus of experiment 'Helping' (Figure 1), but not of experiment 'Cooperation' for two reasons. They would take a lot of space given their size and number, and additionally Figure 2 shows a drawing of one of these apparatuses. We decided on this compromise because that way the paper will not become too cluttered while the reader can still get a basic understanding of how the apparatuses are constructed.

I. 44 Animal instead of primate

→ Please see line 44

I. 67 See also the recent results positive on African Grey parrots of Brucks & von Bayern (2020) *Curr. Biol.*, which use an exact copy of a paradigm used by Massen et al. 2015 *Frontiers in Comp Psychol*, which failed to show any prosocial motivation in ravens.

→ Please see changes made to the first paragraph of the paper

I.84 see also Schwab et al. 2012 *PLoS One*

→ We included the citation, please see line 98.

I. 294 rephrase, now the subject of the sentence changes halfway.

→ Thank you, this sentence is indeed difficult to follow. Please see line 309:

“We tested each ape with two different partners. In one dyad the ape was assigned the role of the helper and in the other dyad the role of the receiver (see Table 1).”

I. 365. as asked before. Which data was used for subsequent analyses? that of the original coder or that of the second coder in those cases where they did not agree?

→ We continued to use the data of the original coder, please see line 379 where we added this information.

I. 604 Maybe add a figure 6b in which you show the same results but now for when the incorrect tool was transferred as to truly contrast them against each other.

→ To keep the paper as concise as possible, we would prefer not to include this figure.

I. 614 While we found MULTIPLE spontaneous...(I think this would stress the difference even more between the bonobos (a lot of transfers) and the chimps (only two))

→ That is true, please see line 624.

Thanks again!

Sincerely,

Suska Nolte & Josep Call

Appendix D

Dear Reviewer 3,

Thank you for taking the time to read the manuscript, inquiring about details, and for your feedback. Please see below our comments and answers to your questions.

“I read the manuscript and also the correspondence between the authors and the reviewers in the previous review round. I acknowledge that the empirical data on chimpanzees’ and bonobos’ helping and cooperation is very important and worth publication in the current situation with still little prior research; however, I also agree with the reviewers’ concerns.

As the reviewers pointed out, the biggest limitation of this study is the small sample size and the imbalance of the participants’ sex ratio between the chimpanzees and bonobos. I understand this often takes place in empirical studies with great apes, but this actually makes it very difficult to generalize the results and discuss the chimpanzee-bonobo species differences. One of the simplest interpretation of the results may be that in mixed-sex pairs, both in chimpanzees and bonobos, individuals are reluctant to hand tool to their partner of different sex. This is irrelevant to the species difference. Thus the authors should tone down their claim of species difference. They should wait for further empirical studies with larger samples or meta-analyses.”

- ➔ We indeed did not observe transfers in any of the mixed-sex dyads, however we included the same amount of mixed-sex dyads in both species (two pairs of either species). The difference between the two species lies in the number of same-sex pairs that we could test. While we tested two female-female and two male-male chimpanzee pairs, due to logistical reasons, we could not test male-male bonobo pairs. Instead, we could only test four female-female bonobo pairs. Based on previous research, the lowest number of transfers would be expected between bonobo males. In contrast, the highest cooperation rates would be expected between chimpanzee males. However, in addition to no transfers between mixed-sex partners we also did not find a single transfer in either of the male-male chimpanzee pairs. Thus, if anything, we could infer that transfers occur between females regardless of the species. However, given that we observed only two instances of transfers between chimpanzee females and it occurred from a mother to her daughter, we do think that the tested chimpanzees behaved different to the bonobos in our task. We strongly agree, however, that more research is needed with larger samples to draw more generalizable conclusions regarding species differences. Toning down our conclusions is something that we did multiple times in the manuscript following the reviewers’ advice, see lines 676 to 684, 704 to 706, and 776 to 778.

“Instead, I suggest that the authors focus more specifically on the bonobos’ data with more detailed analyses. This report should be important since there are still few studies on bonobos’ helping. And it seems interesting that the bonobos are helpful but their helping is not corresponding to their partner’s need, which can be discussed in contrast with the previous chimpanzees’ targeted helping.”

- ➔ We completely agree with the reviewer, which is why we focused the main analyses on the bonobo data aiming to understand which factors influenced whether bonobos transferred tools (i.e. line 388), whether they tailored their help to the needs of the other (i.e. line 415), and whether the receivers approached the helper in order to obtain a tool (i.e. line 483). The only

analyses for which we included the chimpanzee data was to assess whether the two species differed with regard to the receiver's behavior to understand whether the difference in results might be explained by a difference in approach behaviours (i.e. line 457). We propose a follow up study that could be used to delve further into the topic of understanding the partner's needs and differentiating them from the own given that the incentive to pay attention might have been very low in our study (i.e. line 695 to 697).

"I also wonder how the authors decided the pairing of the individuals. Existing social relationships should have influenced the results. To avoid this arbitrary pairing (as well as to increase the sample size), there should have been a possibility to make all the possible pairings in theory (6*5 pairs respectively in chimpanzees and bonobos). Why didn't the authors design like this?"

→ Even though we would have liked to do so, we could not test all possible combinations due to the complexity of the design and time that it took to test each pair (i.e. for each dyad: 12 days per three conditions per three experiments plus several training sessions per experiment, see Figure 3). The social relationship could influence the transfer frequency in the helping task, however, it should not lead to zero transfers in the cooperation task where helpers would gain immediate benefits from tool transfers. Chimpanzees (which are the species that helped the least in the current study), have been shown to cooperate in previous studies with multiple partners provided partners tolerate each other in proximity and the food is divisible. Food was clearly divisible in our study as each partner was located in a different room, thus eliminating direct competition for the spoils. Moreover, spatial separation should also eliminate the concern about lack of tolerance between partners, but even so, we know that all individuals in this study at the very least tolerated being near each other even when present in the same room. In fact, Amici et al. (2012, see Figure 1) observed high levels of tolerance in these two groups in an unrelated study. However, the effect of relationship quality would indeed be interesting to investigate and future studies could include relationship quality measures if they have a sample size that is large enough to draw inferences. To replicate our study (with all its conditions) with many more pairs is important but it would have required a major undertaking which we couldn't do because we did not have enough bonobo pairs or the time to complete all conditions with all pairs.

Amici, F., Call, J., Aureli, F. (2014). Aversion to violation of expectations of food distribution: the role of social tolerance and relative dominance in seven primate species. *Behaviour*, 149, 345-368, <http://dx.doi.org/10.1163/156853912X637833>.

"The other difficult point to interpret is that the chimpanzees practically never demonstrated helping and cooperation. This contradicts the results of the previous studies, and the authors' explanation is not so appealing. There should be many other possible factors that may have affected the results, such as the participants' experience in previous experiments (especially in helping tasks, if they have participated), sex, social relationship between the paired individuals, interaction, explicitness of request behaviors etc. With these results, it is too early to conclude that chimpanzees are reluctant to help others. This should be context-dependent. This also makes the species comparison difficult. As the present and previous studies suggested, explicit demonstration of request behavior should be important

for the apes' helping. Is there any index about the explicitness of request behaviors? Can the authors discuss differences of explicitness in request behaviors between the chimpanzees and bonobos, and also between the present and previous studies?"

→ All of the apes at the MPI participate in cognitive and behavioural tests on a regular basis and all individuals tested have a long history of participating in them, including tasks that involve cooperation, tolerance, and helping behaviour. To our knowledge there was no test that investigated tool transfers earlier than five years ago. Please see above our response and interpretation regarding the sex combination and social relationships. We coded several behaviours (i.e. proximity, reaching, banging, scratching) to assess whether the two partners interacted with each other and the level of stress of the receiver. We did not observe any additional distinct behaviours that might have obviously influenced the likelihood of tool transfers and would have reported them otherwise. We agree that request behaviour most likely influences transfers and assessed whether receivers requested tools and whether the two species differed in this regard. Given the lack of opportunity to use the mesh of the testing rooms differently, the apes could not show variation in requesting behaviours and could only resort to either banging against or sticking the fingers through the mesh. Thus, neither species could be more or less explicit in their requesting behaviours, and we would have reported any unanticipated deviation from it. For both species, the request behaviour was scored in the same way and our interrater reliability was high. Still, we did not find a species difference in the likelihood to request for tools (see line 594). Thus, while we found that request behaviours did influence the likelihood that bonobo helpers shared a tool (see line 564), the lack of transfers in chimpanzees could not be explained by request behaviours given the same frequency of requesting. We discuss differences of explicitness in request behaviours and the possible influence on helping behaviour on line 713 to 718.

"I did not find any captions for the figures."

→ We are sorry you faced this problem and copied all figures with captions below, so that you have them readily available.

Figure 1 Picture of ‘apparatus Stick’ (A) and ‘apparatus Block’ (B). The flat surface was attached to the testing cage and faced the ape. Red numbers indicate the location where the tool needed to be inserted (1), the location of the grapes (2), the slide (3), and the feeding area (4). The red error respectively indicates the direction of movement of the container or lever.

Figure 2 Set-up of experiment ‘Helping’ (A) and experiment ‘Cooperation’ (B), depicting the condition ‘Different’. The same set-up was used in condition ‘Same’ and the control condition. The tools were located underneath the grey box and a second box was used in the control condition.

Figure 3 Flow-chart of the entire study design.

Figure 4 Number of sessions with at least one tool transfer in each of the three conditions for each experiment (A) and for each dyad across all experiments (B). Across all three experiments, every dyad received a total of 36 sessions in each condition. Pair “Yasa-Fimi” is a mother-infant dyad.

Figure 5 Proportion of transfers across the three conditions within each session separately plotted for experiment ‘Helping 1’ (A), ‘Cooperation’ (B), and ‘Helping 2’ (C). Proportions being 0 denote helpers that did not share a tool in any of the three conditions. Proportions being 0.33 denote helpers that only shared in one condition on that session. Similarly, proportions being 0.66 denote helpers that shared a tool in two conditions on that session. Finally, proportions being 1 denote helpers that shared a tool on all three conditions on that session. The larger the area of the points, the more helpers acted in such a manner on the same session. The interaction of session and experiment can be visually assessed as transfers were done randomly across sessions in experiment ‘Helping 1’ and ‘Helping 2’ but increase to a consistent level around session 6 in experiment ‘Cooperation’.

Figure 6 Proportion of sessions in which the receivers put their fingers through the mesh (reaching), remained close to the mesh dividing the two cages (close proximity), and scratched themselves (scratching) before and after the correct tool was shared while controlling for the influence of the other predictors in the model.

Appendix E

Dear Editor and Reviewers,

Editor comments:

Thank you for your efforts at revising and resubmitting. One reviewer is happy at this point but the other continues to have serious concerns. These need to be answered very conscientiously, please, in your final revision. Note that we will send the MS to this reviewer for final comment. If issues still remain we will unfortunately not be able to consider the manuscript further. Thanks again.

Response: Thank you very much for giving us the opportunity to revise our manuscript.

Reviewer comments to Author:

Reviewer: 1

Comments to the Author(s)

I thank the authors for considering my comments.

Response: Thank you.

Reviewer: 3

Comments to the Author(s)

I read the authors' replies to my previous review comments, and found that they are clear except one point. Though my concerns about the small sample size and the selection of pairs still remain, I also understand the restriction.

I would like to raise a few more comments which may hopefully be helpful to improve the manuscript.

Response: Thank you very much for your thoughts and feedback. We addressed each aspect in detail below and in our manuscript. We hope that our responses will satisfactorily address your concerns.

1. The authors say "The social relationship could influence the transfer frequency in the helping task, however, it should not lead to zero transfers in the cooperation task where helpers would gain immediate benefits from tool transfers." In their reply to my review comment, but I don't agree with this. Even in humans, people do not cooperate with a partner in bad relationship even when it is mutually beneficial contexts (see politics everywhere in the world...).

Response: This is true, but note that humans are much more irrational than chimpanzees and bonobos in this regard. For instance, in the ultimatum game, humans recipients do forego a reward to punish an unequal offer (e.g., Gueth et al., 1982; Heinrich et al., 2005), but chimpanzee and bonobo recipients do

not (Jensen et al., 2007; Kaiser et al., 2012; Proctor et al., 2013). Still, we are fully aware that tolerance (and a positive relationship in some studies) is one of the key factor determining whether chimpanzees will cooperate with or help each other (e.g., Melis et al., 2006). This is why we considered this aspect carefully when we selected who would be paired with whom in our study. See below please.

In the Methods section, the authors should mention more clearly how they selected the pairs from the larger pool of possible candidates (how many chimpanzees and bonobos were there in MPI?) and why they didn't test the others.

Response: Thank you for pointing out that our method for selecting the pairs was missing. We used two indicators to determine this. First, we included those pairs who had participated reliably in previous tests of cooperation (e.g., Sánchez-Amaro et al., 2016). Second, we followed the keepers' advice about the pairs that were tolerant with each other and who had a good relationship based on the keepers' experience regarding the apes' grooming and spatial closeness habits. We have now included a detailed explanation of our selection criteria in the method section (lines 234-241). Moreover, despite its importance, we have discussed why we think that relationship quality cannot solely explain the absence of cooperation of all chimpanzee dyads in our experiment (lines 768-783).

2. When discussing cooperation, especially prosociality, among wild chimpanzees and bonobos, it is more relevant to discuss food sharing, a typical dyadic prosocial behavior in the wild. There have been several studies both in chimpanzees and bonobos, which would enhance the discussion on species comparison (e.g. males share food with others in chimpanzees while females share food with others in bonobos).

Response: We have made this aspect more explicit in our manuscript, which entailed revising parts of the introduction and discussion accordingly (lines 115-123 and 759 - 768).

3. There should be some other interpretation on the result that the bonobos did not select appropriate tools according to the partner's need. One plausible explanation might be that the apparatus was complicated and not so intuitive, and it might have been difficult for a helper to see which tool is needed on the partner's side. Though they could "understand" how the apparatus works, this ability may be exercised for his/her own reward, but not for the others when it requires some cognitive load.

Response: Yes, that is true, and a good alternative explanation. We have included it in the discussion (lines 698-702 and 705).

4. The chimpanzees and the bonobos were tested in the same experimental rooms, weren't they? The exact size of the rooms is informative.

Response: Not the same room, but two separate rooms with nearly identical dimensions and the same layout. We have included this info in the method section (lines 304-307) and additionally we included detailed pictures of the testing compartments and explanations in the supplementary material.

Again, thanks a lot for your input and helping to improve this manuscript.

Kind regards,

Nolte & Call

References

Gueth, W., Schmittberger, R. & Schwarze, B. (1982). An experimental analysis of ultimatum bargaining. *J. Econ. Behav. Org.* 3, 367-388.

Henrich, J. et al. (2005). 'Economic man' in cross-cultural perspective: behavioral experiments in 15 small-scale societies. *Behav. Brain Sci.* 28, 795-815.

Jensen, K., Call, J., & Tomasello, M. (2007). Chimpanzees are rational maximizers in an ultimatum game. *Science*, 318, 107-109.

Kaiser, I., Jensen, K., Call, J., & Tomasello, M. (2012). Theft in an ultimatum game: chimpanzees and bonobos are insensitive to unfairness. *Biology Letters*, 8, 942-945.

Melis, A.P., Hare, B., & Tomasello, M. (2006). Engineering cooperation in chimpanzees: tolerance constraints on cooperation. *Animal Behaviour*, 72, 275-286.

Proctor, D., Williamson, R. A., de Waal, F. B. M., & Brosnan, S. F. (2013). Chimpanzees play the ultimatum game. *Proceedings of the National Academy of Sciences*, 110, 2070-2075. doi:10.1073/pnas.1220806110

Sánchez-Amaro, A., Duguid, S., Call, J., & Tomasello, M. (2016). Chimpanzees coordinate in a snowdrift game. *Animal Behaviour*, 116, 61-74